# Multi-BK-Net: Multi-Branch Multi-Kernel Convolutional Neural Networks for Clinical EEG Analysis

**Ann-Kathrin Kiessner**                                       *kiessner@cs.uni-freiburg.de*
*Department of Computer Science & IMBIT//BrainLinks-BrainTools, University of Freiburg*

**Joschka Boedecker**                                           *jboedeck@cs.uni-freiburg.de*
*Department of Computer Science & IMBIT//BrainLinks-BrainTools, University of Freiburg & Collaborative Research Institute Intelligent Oncology (CRIION)*

**Tonio Ball**                                              *tonio.ball@uniklinik-freiburg.de*
*Neuromedical AI Lab, Medical Centre Freiburg & IMBIT//BrainLinks-BrainTools, University of Freiburg*

**Reviewed on OpenReview:** *https://openreview.net/forum?id=IsG1OxZAaA*

## Abstract

Classifying an electroencephalography (EEG) recording as pathological or non-pathological is an important first step in diagnosing and managing neurological diseases and disorders. As manual EEG classification is costly, time-consuming and requires highly trained experts, deep learning methods for automated classification of general EEG pathology offer a promising option to assist clinicians in screening EEGs. Convolutional neural networks (CNNs) are well-suited for classifying pathological EEG signals due to their ability to perform end-to-end learning. In practice, however, current CNN solutions suffer from limited classification performance due to I) a single-scale network design that cannot fully capture the high intra- and inter-subject variability of the EEG signal, the diversity of the data, and the heterogeneity of pathological EEG patterns and II) the small size and limited diversity of the dataset commonly used to train and evaluate the networks. These challenges result in a low sensitivity score and a performance drop on more diverse patient populations, further hindering their reliability for real-world applications. Here, we propose a novel multi-branch, multi-scale CNN called Multi-BK-Net (Multi-Branch Multi-Kernel Network), comprising five parallel branches that incorporate temporal convolution, spatial convolution, and pooling layers, with temporal kernel sizes defined by five clinically relevant frequency bands in its first block. Evaluation is based on two public datasets with predefined test sets: the Temple University Hospital (TUH) Abnormal EEG Corpus and the TUH Abnormal Expansion Balanced EEG Corpus. Our Multi-BK-Net outperforms five baseline architectures and state-of-the-art end-to-end approaches in terms of accuracy and sensitivity on these datasets, setting a new benchmark. Furthermore, ablation experiments highlight the importance of the multi-branch, multi-scale input block of the Multi-BK-Net. Overall, our findings indicate the efficacy of multi-branch, multi-scale CNNs in accurately and reliably classifying EEG pathology, demonstrating advantages in handling data heterogeneity compared to other deep learning approaches. Thus, this study contributes to the ongoing development of deep end-to-end methods for general EEG pathology classification.

# 1 Introduction

Electroencephalography (EEG) is a non-invasive method of measuring and recording electrical activity in the brain. EEG has high temporal resolution, low equipment cost, and high sensitivity to dynamic changes in neural signals. Due to its high efficiency and usability, EEG is most commonly used in clinical practice for the diagnosis and management of various neurological conditions (Lopez et al., 2015; Zhang et al., 2023b). For this purpose, a preliminary important step in clinical practice is to classify an EEG recording as non-pathological or pathological (Brogger et al., 2018; Lopez et al., 2015). To this end, human EEG experts visually analyse recordings from long-term monitoring or multiple short sessions, which is a tedious and time-consuming process (Brogger et al., 2018). They also consider various additional patient-related factors, such as medical history, age, or medication (Beuchat et al., 2021; Nayak & Anilkumar, 2020). Furthermore, it requires years of training to achieve a thorough understanding of pathological EEG patterns and to distinguish them from normal EEG, normal benign variants, and artefacts (Amin et al., 2023; Emmady & Anilkumar, 2023; Hoppe, 2018). Thus, these challenges result in inter-rater variability and diagnostic errors. For classifying EEG recordings as pathological or non-pathological, previous research has reported inter-rater agreement of 86–88% between two neurologists (Houfek & Ellingson, 1959; Rose et al., 1973). At the same time, Beuchat et al. (2021) found even lower inter-rater agreement, 82-86%, among multiple EEG technologists and neurologists.

In this sense, introducing automated EEG classification, or at least some level of automation such as clinical decision support systems, has the potential to improve or accelerate the EEG classification process, thereby enhancing the quality of patient care. To this end, deep learning approaches for EEG classification have received increasing attention in recent years (for a detailed review, see Amrani et al., 2021; Craik et al., 2019; Faust et al., 2018; Rahman et al., 2024; Roy et al., 2019b; Praveena et al., 2022). It has also spurred their application to the task of classifying general EEG pathology (Schirrmeister et al., 2017b). In this regard, research has shown that convolutional neural networks (CNNs) have been quite effective for general EEG pathology classification due to their ability to extract relevant feature representations directly from raw or minimally preprocessed EEG data (Darvishi-Bayazi et al., 2023; Gemein et al., 2020; Khan et al., 2022; Kiessner et al., 2023; 2024; Van Leeuwen et al., 2019; Western et al., 2021, Appendix A.1 provides more details on the related works.). In practice, however, the classification performance of current state-of-the-art CNN methods is primarily limited by three factors. First, the EEG presents inherent challenges which make the application of deep learning methods to real-world EEG datasets more difficult. For example, the recorded EEG signals are high-dimensional, non-linear, nonstationary (Cole & Voytek, 2019; Gramfort et al., 2013; Jia et al., 2021), have a low signal-to-noise ratio (Bigdely-Shamlo et al., 2015; Jas et al., 2017), are strongly influenced by artefacts caused by external environmental factors (e.g. electrical interference from external sources) (Islam et al., 2016; Kane et al., 2017) or physiological sources (e.g. cardiac, muscle activity, eye movements, or fatigue) (Britton et al., 2016), and variations in recording protocols and labelling standards within clinical data can occur (Poziomska et al., 2025). In addition, pathological EEG patterns exhibit a wide range of physiological variability across both patients (Nahmias et al., 2019) and neurological conditions (Emmady & Anilkumar, 2023; Nayak & Anilkumar, 2020; Smith, 2005). Due to these challenges, the classification performance can vary between patients (Altuwaijri et al., 2022; Lashgari et al., 2020; Roy et al., 2019b; Schirrmeister et al., 2017b). Second, due to the scarcity of publicly available datasets, current approaches are mainly trained and evaluated on a single dataset, the TUH Abnormal EEG Corpus (TUAB) (López de Diego, 2017). While it is therefore well established that these methods perform well on this small, homogeneous dataset, this limitation restricts the classification performance of these approaches. For example, recent studies have observed significant differences in the performance of several previously established models after training or validating them on other datasets (Darvishi-Bayazi et al., 2023; Khan et al., 2022; Kiessner et al., 2023; 2024; Nahmias & Kontson, 2020; Poziomska et al., 2025; Van Leeuwen et al., 2019; Western et al., 2021). Lastly, single-scale convolutions, with manually and arbitrarily defined kernel sizes (Emsawas et al., 2022), are primarily used to design CNNs; however, they are less capable of handling individual differences in the EEG signal. For example, several studies have found that, due to variability in the EEG signal, the optimal kernel size differs from subject to subject and from time to time for the same subject (Altaheri et al., 2023b; Altuwaijri et al., 2022; Jia et al., 2021). While multi-branch,

multi-scale[1], or parallel architectures (Altuwaijri et al., 2022; Belwafi et al., 2017; Ingolfsson et al., 2020; Jia et al., 2021; Riyad et al., 2020; Szegedy et al., 2015; Zhang et al., 2023a) have shown promising results in addressing the challenge of heterogeneity in various EEG classification tasks (Cai et al., 2024; Jia et al., 2021; Siddiqa et al., 2024; Yan et al., 2025; Zhu et al., 2023), only a few attempts have been made to address the challenges of general EEG pathology classification by using CNNs with a set of three convolution scales (Brenner et al., 2024; Roy et al., 2019a; Wu et al., 2021). However, these CNN approaches have achieved classification performance similar to that of other CNNs, with accuracies ranging from 85.10% to 87.10%; they also inherit the disadvantages found in their single-scale counterparts. For example, these results were based on the TUAB, and no attempts have been made to evaluate the performance of these CNNs on a larger, more heterogeneous dataset. In addition, smaller kernel sizes are preferred due to lower computational costs (Emsawas et al., 2022), which, however, tend to learn shorter temporal patterns from faster frequency bands (Cohen, 2014; Jia et al., 2021). This also restricts the ability of these CNNs to capture the heterogeneity of the EEG signal (Altaheri et al., 2023b; Emsawas et al., 2022). Therefore, developing an accurate and reliable CNN solution for general EEG pathology classification requires a network design that addresses the high intra- and inter-subject variability in the EEG signal, data diversity, and the heterogeneity of pathological EEG patterns.

In this work, we propose a novel, multi-branch, multi-scale CNN, called Multi-Branch Multi-Kernel Network (Multi-BK-Net) for general EEG pathology classification that extracts long-term and short-term spatiotemporal EEG features by incorporating five parallel branches within the first convolution-pooling block (see Section 2.1). Each branch employs a temporal convolution layer with a different kernel size that was defined considering five clinically relevant frequency bands, namely delta (1–3 Hz), theta (4–7 Hz), alpha (8–13 Hz), beta (14–30 Hz) and low gamma (30-80 Hz) (Brenner et al., 2024). Aiming at the problem of a small amount of training data and to increase data diversity, we combined the predefined training sets of two publicly available datasets for general EEG pathology classification, the TUH Abnormal EEG Corpus (TUAB) (López de Diego, 2017) and TUH Abnormal Expansion Balanced EEG Corpus (TUABEXB) (Kiessner et al., 2023), to optimise and train our Multi-BK-Net. The hyperparameters of the Multi-BK-Net were optimised using multivariate tree-structured Parzen estimators (TPE) (Bergstra et al., 2011; 2013) from the Optuna library (Akiba et al., 2019) with respect to the mean validation accuracy and mean validation sensitivity values from a 5-fold cross-validation. Our study presents the following contributions to research on deep end-to-end CNNs for general EEG pathology classification:

- We propose the Multi-BK-Net, a multi-branch, multi-scale CNN designed for general EEG pathology classification. We combine the concepts of multi-scale and multi-branch CNNs with the idea of adapting the temporal kernel size based on five clinically relevant frequency bands to address the challenge of heterogeneity inherent in the EEG signal.

- We evaluate our proposed Multi-BK-Net on the predefined test sets of two publicly available datasets, the TUAB and the TUABEXB datasets. Our Multi-BK-Net achieves mean accuracies of 87.75% and 87.01% and mean sensitivities of 83.10% and 84.25%, respectively, and thus outperforms five architectures used as a baseline (Section 3.4), as well as previously reported state-of-the-art deep learning approaches (Section 3.5) on these datasets, setting a new benchmark. In addition, with mean accuracies in the narrow range of 87–88%, the Multi-BK-Net approaches the upper limit of inter-rater agreement among human experts, thus bringing deep learning-based general EEG pathology classification closer to human-level performance. An improvement in evaluation sensitivity demonstrates further practical gains of our method. In comparison to the baseline architectures, Multi-BK-Net achieved the highest mean sensitivities at a fixed specificity of 95% on both test sets (82.78% and 84.25%). The significance of this result lies in applications such as deep learning-based EEG classification and clinical decision support systems in clinical practice, for which a high, robust sensitivity is crucial for accurately identifying EEG pathology. In deep learning-based clinical EEG analysis, high accuracy and sensitivity are crucial for improved patient outcomes, reduced diagnostic errors, and enhanced clinical efficiency. Thus, even minor gains significantly improve the reliability and trustworthiness of such systems.

---

[1]In this work, we will use "multi-scale" and "multi-kernel" interchangeably.

- Our ablation experiments show that using both multiple temporal kernel lengths and multiple parallel branches in the first block significantly improves the classification performance of the model, particularly in terms of mean accuracy and mean sensitivity (Section 3.6).

- To interpret the trained Multi-BK-Net, we visualise the learned features using Uniform Manifold Approximation and Projection (UMAP) (McInnes et al., 2018) method (Section 3.7) and observe that the Multi-BK-Net forms distinct and compact clusters of samples from the pathological and non-pathological classes, which partly explains the good performance of this architecture. Additionally, we perform an amplitude gradient analysis (Section 3.8) that shows that the model's pathological prediction is sensitive to localised patterns of amplitude changes across different frequency bands. These patterns align with the current neurophysiological knowledge for pathological EEG patterns (Amin et al., 2023; Emmady & Anilkumar, 2023; Hoppe, 2018; Kane et al., 2017; Medithe & Nelakuditi, 2016; Tatum & William, 2021) and with the pathological patterns that have been identified by human experts in the corresponding clinical EEG reports.

- A qualitative review of clinical EEG reports corresponding to false negative and false positive classified EEG recordings by the Multi-BK-Net suggests that the CNN struggles to identify context-dependent abnormalities. While clinicians incorporate additional information about the patient's age, vigilance, medications, and clinical history to interpret EEG patterns, our CNN, which was trained only on the EEG signal, lacks access to this contextual information. This results in the model being at a disadvantage relative to clinicians and highlights the potential for further performance gains by incorporating contextual information into the training and classification processes.

- Overall, our results demonstrate the efficacy of our multi-branch, multi-scale CNN solution in accurately and reliably classifying general EEG pathology. Our findings contribute to the broader discussion on the challenges of deep learning-based general EEG pathology classification, providing new insights for the design of future end-to-end CNNs. This will ultimately advance the development of more reliable and robust methods for deep learning-based general EEG pathology classification.

## 2 Methods

### 2.1 Multi-Branch Multi-Kernel Network (Multi-BK-Net)

We propose a novel deep, multi-scale, and multi-branch CNN, called Multi-BK-Net, for general EEG pathology classification. This model extracts long-term and short-term spatiotemporal EEG features by incorporating a dedicated multi-branch, multi-scale first block that processes EEG inputs for different temporal patterns. The block is followed by three standard convolution-mean-pooling blocks and a softmax classification layer. Figure 1 shows the overall structure of the proposed Multi-BK-Net architecture. The first convolutional block is divided into five branches (multi-branches). Each convolutional branch contains two convolutional layers and a mean-pooling layer. The first convolutional layer performs temporal filtering with a set of seven convolutional filters. Each branch uses a different kernel size for temporal filtering (multi-scale). This allows the model to extract and combine features from different time scales. The choice of kernel size was based on five frequency bands, namely, delta (0.5 to 4 Hz), theta (4 to 7 Hz), alpha (8 to 12 Hz), beta (13 to 30 Hz), and low gamma (30 to 80 Hz) (Brenner et al., 2024; Nayak & Anilkumar, 2020), by also considering the sampling rate. In this study, the EEG signal is represented as a 2D input, with the dimension ($E \times T$) and with a window size of 60 seconds and a sampling frequency of 100 Hz, corresponding to 6000 time points, an input shape of $21 \times 6000$ (for more details, see Section 3.1). Accordingly, we used five different temporal kernel sizes: $1 \times 200$, $1 \times 25$, $1 \times 13$, $1 \times 7$ and $1 \times 3$. In the second layer, a spatial convolution is applied. Each filter performs a spatial convolution with weights for all possible electrode pairs, using the filters from the preceding temporal convolution. Note that there is no activation function between the two layers; therefore, in principle, they could be combined into a single layer. However, we chose a split convolution because it has been shown to outperform a combined temporal-spatial convolutional layer (Schirrmeister et al., 2017a;b). After the spatial convolution, mean pooling is applied. As successfully done by (Schirrmeister et al., 2017a), the pooling strides were moved directly to the convolutional layers preceding each pooling. Finally, the outputs of each branch are concatenated along the feature axis, resulting in a

Table 1: Hyperparameters of the Multi-BK-Net architecture. Hyperparameters have been optimised in preliminary experiments; more details on the hyperparameter optimisation and design choices are provided in Appendix A.3.

| Hyperparameter | Selected value |
|---|---|
| Total number of temporal convolution filters | 35 (7 filters per branch) |
| Normalisation | GroupNorm |
| Activation functions | GELU |
| Pooling mode first block | Mean |
| Pooling mode remaining blocks | Mean |
| Forth conv-pooling-block | True |
| Forth conv-pooling-block broader | True |
| Pool length | 3 |
| Pool stride | 3 |
| Stride before pool | True |
| Dropout | 0.502959339666169 |
| Filter length convolution blocks | 20 |
| Input window size | 6000 |
| Weighted loss factor pathological | 1 |
| Optimizer | AdamW |
| Optimizer beta1 | 0.5 |
| Learning rate | 0.0031414364096615 |
| Weight decay | 1.8397405899531204e-05 |
| Batch size | 64 |
| Number of epochs | 42 |
| Number of channels | 21 |

feature vector of shape (1, 35, 397, 1). The concatenated tensors are then passed through three convolution-mean-pooling blocks. Each convolution-mean-pooling block consists of a convolutional layer and a mean pooling layer. The output of the fourth block is then passed through a softmax classification layer to receive the final prediction. In total, the Multi-BK-Net contains 1,038,683 trainable parameters. We implemented our model in Braindecode (BD), an open-source Python toolbox for decoding raw electrophysiological brain data with deep learning models (Schirrmeister et al., 2017b). The Xavier initialisation method (Glorot & Bengio, 2010) was used to initialise the weights. The biases have been initialised to 0. We applied group normalisation (Wu & He, 2018) to the output of the convolutional layers before the nonlinearity. We used Gaussian Error Linear Units, or GELUs ($GELU(x) = xP(X \leq x) = x\Phi(x)$) (Hendrycks & Gimpel, 2023) as activation functions. Additionally, we used dropout as a regularisation strategy (Srivastava et al., 2014). Appendix A.3 provides a more detailed description of the design choices and hyperparameter optimisation procedure.

## 2.2 Network Training

For optimisation, we used the adaptive moment estimation with decoupled weight decay (AdamW) optimiser (Loshchilov & Hutter, 2019) with an optimised learning rate and beta1 parameter, and the negative log-likelihood loss (Terven et al., 2024). We used cosine annealing (Loshchilov & Hutter, 2016) to schedule the learning rates for both the gradient and weight decay updates, and refrained from restarting the learning rate. We trained the model on non-overlapping, equally sized time windows of size 60 seconds using the trial-wise training strategy[2] as described by Schirrmeister et al. (2017b). We repeated the training 10 times, using a fixed set of random seeds to account for model variance arising from random weight initialisation. The final predictions are obtained by first averaging the predictions of each window for each recording,

---

[2]During trial-wise training, a complete window is pushed through the network. The network then produces a prediction, which is compared to the target (label) for that window (trial) to compute the loss.

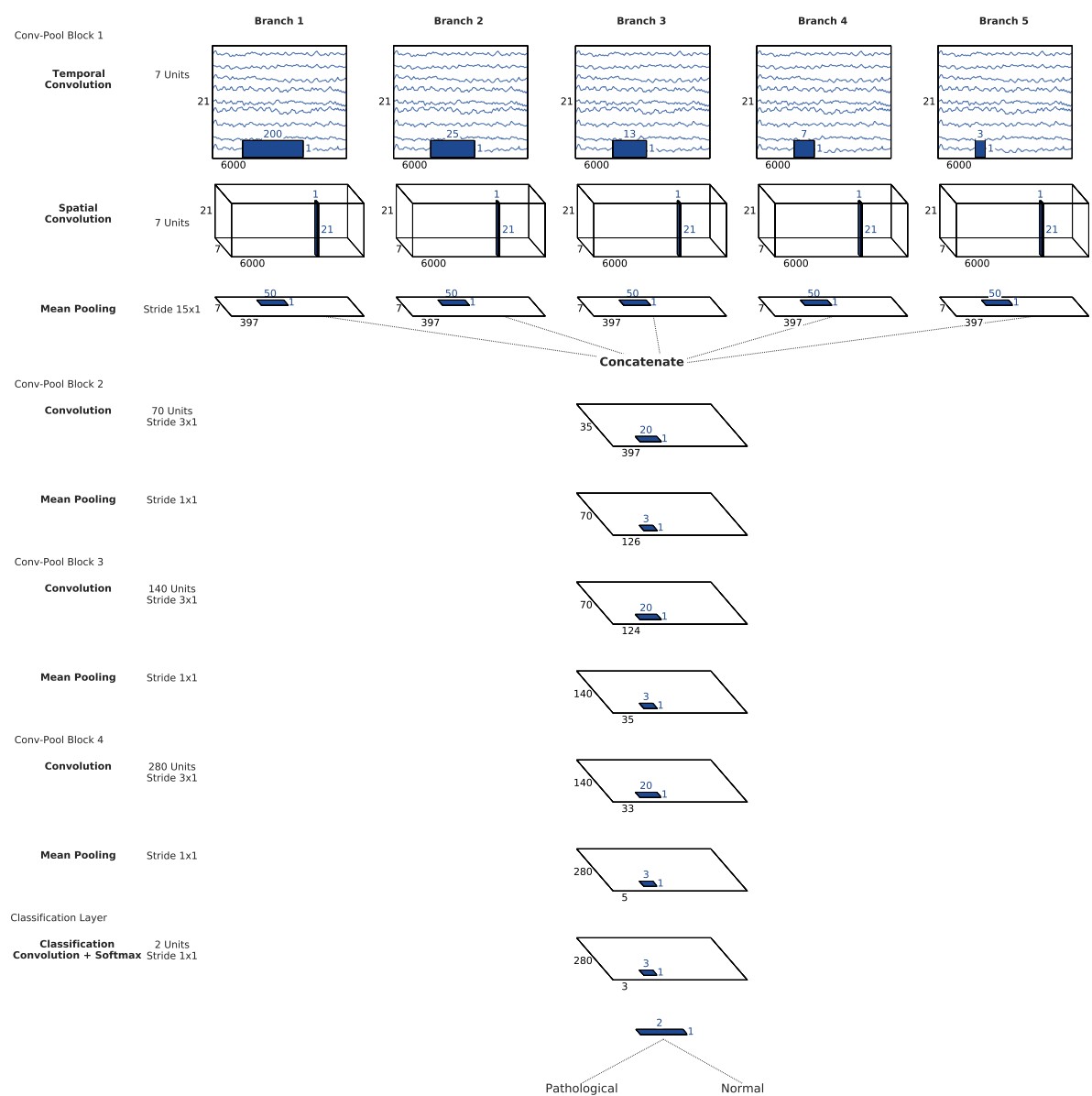

Figure 1: Multi-BK-Net architecture. The first block is divided into five branches, each applying a temporal convolution with a different kernel size, followed by a spatial convolution. There is no activation function between the two convolutional layers; after that, a mean-pooling layer is added. The EEG input (at the top) is passed through each branch. The outputs of all branches are concatenated and then used as input for the subsequent convolution-mean-pooling blocks. Black cuboids: input/feature maps; coloured cuboids: convolution/pooling kernels. The corresponding sizes are shown in black or in colour. Note that the proportions of maps and kernels in this schematic are only approximate.

and then calculating the overall performance across recordings (see Section 3.3 for more details). The list of hyperparameters for training is provided in Table 1. The code of this study is available at `https://github.com/nrgrp/Multi-BK-Net-general-EEG-pathology-classification.git`.

## 3 Experiments and Results

To demonstrate the efficacy of our method, we evaluated the Multi-BK-Net on two public datasets for general EEG pathology classification: I) the small, homogeneous, public dataset TUH Abnormal EEG Corpus (TUAB) (López de Diego, 2017) for comparison with previously published accuracies and II) the larger, recently introduced TUH Abnormal Expansion Balanced EEG Corpus (TUABEXB) (Kiessner et al., 2023) to evaluate the classification methods with a larger heterogeneous number of EEG recordings. First, we compared the performance of the Multi-BK-Net to five baseline architectures. Following this, we conducted a performance comparison with previously reported state-of-the-art deep learning approaches. We then performed ablation experiments to demonstrate the superiority of the proposed Multi-BK-Net, further highlighting the importance of using multiple temporal kernel sizes and multiple branches in the first block. In addition, we visualised the learned features of the Multi-BK-Net to highlight its robustness and provide a comprehensive understanding of model performance. Lastly, we performed an amplitude gradient study to determine how sensitive the Multi-BK-Net's predictions are to amplitude changes across different frequency bands.

### 3.1 Datasets and Preprocessing Steps

The TUAB consists of 2,993 recordings (49.18% pathological) from 2,329 patients (52.09% female, mean age: $48.55 \pm 17.86$ years) that are divided into a predefined training set (2,717 recordings) and an evaluation set (276 recordings). In contrast, the TUABEXB contains 8,879 recordings (49.75% pathological) obtained from 7,006 patients (mean age: $47.7 \pm 21.2$ years; 51.7% female) and is divided into a predefined training set (7990 recordings) and an evaluation set (889 recordings). For training, we concatenated the TUAB and TUABEXB training sets, which we refer to as the TUH Abnormal Combined EEG Corpus (TUABCOMB). The TUABCOMB contains 10,707 recordings from 8,549 patients (mean age: $47.91 \pm 20.53$ years; 52.5% female), of which 5,321 recordings (49.70%) are classified as pathological. We evaluated our models on the predefined test sets of the TUAB and TUABEXB datasets, respectively (for more details on the datasets, see Section 3.3). There is no overlap of patients between the TUAB and the TUABEXB datasets. In both datasets, there is no overlap between patients in the predefined training set and the final evaluation set.

We applied the following minimal preprocessing steps to the raw EEG data: a) selected a set of 21 electrode positions [3] present in all recordings; b) discarded the first 60 seconds of each recording as they contain stronger artefacts; c) used up to 20 minutes of the remaining recording to speed up the computations; d) clipped the EEG recordings at $\pm$ 800 $\mu$V to reduce the effect of strong artefacts; e) re-referenced all recordings to the Common Average Reference (CAR) f) resampled the data to 100 Hz to account for the different sampling rates and to further speed up the computation. To apply CNNs to EEG classification, the raw EEG signals measured in microvolts ($\mu$V) can be represented as input as a 2D array with the number of time steps (T, temporal dimension) as the width and the number of electrode channels (E, spatial dimension) as the height, resulting in an input shape of E×T. To create the inputs for the networks, we process sliding windows over the EEG data $D \in \mathbb{R}^{T \cdot E}$, where $T$ denotes the recording duration and $E$ the number of electrodes. Windows are created as input features $x_t \in R^I$ with input dimensionality $I = E \cdot \lfloor T/S \rfloor \cdot f_s$ at time point $t$, where $S$ is the stride and $f_s$ is the sampling frequency. We use non-overlapping windows of 60 seconds, which corresponds to a stride of $S = 60$ seconds. The corresponding labels are $y \in \{0,1\}^{\lfloor T/S \rfloor}$, where $y_i = 0$ indicates a non-pathological window and $y_i = 1$ indicates a pathological window. A tensor of shape (batch_size, E, T) is provided as input to the architectures.

### 3.2 Baselines

To illustrate the superiority of the proposed Multi-BK-Net, we compare it to several approaches reported in previous work. For general EEG pathology classification, comparing approaches is challenging without evaluating them within the same framework, because factors such as different evaluation methods, preprocessing steps, training strategies, and datasets or dataset versions can introduce bias. Therefore, to evaluate the performance of our architecture, we compared our method with the following five baseline architectures:

---

[3]This set of EEG channels included the channels A1, A2, C3, C4, Cz, F3, F4, F7, F8, Fp1, Fp2, Fz, O1, O2, P3, P4, Pz, T3, T4, T5 and T6.

- Deep4Net (Schirrmeister et al., 2017b;a), a CNN consisting of four convolution-max-pooling blocks.

- ShallowNet (Schirrmeister et al., 2017b;a), a CNN with a convolution-pooling block that was inspired by the Filter Bank Common Spatial Patterns pipeline (Ang et al., 2008; Chin et al., 2009).

- TCN (Bai et al., 2018; Chrabąszcz, 2018; Gemein et al., 2020), a temporal CNN consisting of five residual blocks.

- EEGNet (Lawhern et al., 2018), a compact CNN with depthwise and separable convolutions.

- ChronoNet (Roy et al., 2019a), a deep recurrent neural network combining multiple Conv1D layers with multiple filters of varying sizes and stacked Gated Recurrent Unit (GRU) layers.

These architectures have been implemented in Braindecode and have also demonstrated high performance on TUAB and/or TUABEXB across various studies using different preprocessing steps. Additional details on the baseline architectures, the list of hyperparameters for training, and the training procedure are provided in the Appendix A.4.

## 3.3 Evaluation of Classification Performance

We trained each architecture on the full TUABCOMB training set and then evaluated the models on the withheld final evaluation sets, as predefined in TUAB and TUABEXB, respectively. To manage the statistical variance caused by initialisation and to improve the comparison between training and model configurations, we repeated the training and evaluation ten times for each model and reported the average of evaluation metrics across ten independent runs (Bouthillier et al., 2021; Picard, 2023; Wightman et al., 2021). To ensure comparability with previous work, we evaluated the overall performance of a model using the prediction for each recording. The model's prediction for a recording was calculated by averaging the outputs from all windows of that recording. Each recording was then classified as non-pathological or pathological based on its mean window probability. The performance of the different architectures was evaluated using the following general classification metrics: *accuracy*, *balanced accuracy*, *sensitivity*, *specificity* and *F2-score*. We computed the evaluation metrics for each run and then averaged the results across ten independent runs. In addition, we used the Mann-Whitney U test (Mann & Whitney, 1947) to test for the statistical significance of the difference in performance metrics between the proposed Multi-BK-Net and each of the baseline architectures (H1). The null hypothesis (H0: No significant differences in the performance of Multi-BK-Net compared to a baseline architecture) was rejected with a p-value of $p < 0.05$. To correct for multiple testing, we additionally performed a Bonferroni correction (Bonferroni, 1936) with $\alpha = 0.05$ for all performance comparisons involving our Multi-BK-Net and each baseline architecture on the corresponding evaluation set. Statistical evaluation is based on 10 independent runs. In addition, we calculate the mean Receiver Operating Characteristic (ROC) curve and Precision-Recall (PR) curves. To this end, ROC and PR curves were computed for each run, respectively. The curves from all 10 independent runs were then averaged pointwise to obtain the mean ROC and PR curves. We also calculated the area under the curve (ROC-AUC and PR-AUC) for each run and reported the mean and standard deviation across runs.

## 3.4 Performance Comparision to Baseline Architectures

In this section, we examine the performance of the proposed Multi-BK-Net compared with five baseline architectures on two predefined test datasets, TUAB and TUABEXB. The classification performance of the Multi-BK-Net and all baseline architectures on the TUAB is shown in Figure 2. With a mean accuracy of 87.75%, the Multi-BK-Net shows the best classification performance. This was also statistically significantly better than the baseline models ($p < 0.001$, Mann-Whitney U test). However, in medical diagnosis, such as EEG pathology classification, it is more important to identify all potential pathological cases (high recall/sensitivity) even at the expense of some false positives (lower precision). In this context, the F2-score is a suitable additional metric to compare the different architectures. With a mean sensitivity of 83.10% and a mean F2-score of 84.05%, Multi-BK-Net statistically significantly outperformed all other architectures ($p < 0.001$, Mann-Whitney U test), indicating better recall. Figure 3 compares the classification

Table 2: Mean sensitivity [%] at 95% specificity. Standard deviation is given in parentheses. For each architecture and test set, sensitivity at a fixed specificity of 95% was calculated for each run and then averaged across all ten runs (n=10).

| Architecture | TUAB | TUABEXB |
|---|---|---|
| Deep4Net | 48.97 ($\pm$ 32.52) | 66.99 ($\pm$ 6.80) |
| ShallowNet | 55.87 ($\pm$ 36.60) | 77.90 ($\pm$ 2.37) |
| TCN | 77.94 ($\pm$ 0.59) | 79.16 ($\pm$ 0.79) |
| EEGNet | 70.48 ($\pm$ 2.90) | 52.78 ($\pm$ 34.64) |
| ChronoNet | 78.89 ($\pm$ 0.95) | 40.34 ($\pm$ 40.35) |
| Multi-BK-Net | 82.78 ($\pm$ 1.85) | 84.25 ($\pm$ 0.49) |

performance of the Multi-BK-Net and five baseline architectures on the TUABEXB evaluation set. Again, the Multi-BK-Net outperformed the baseline models. In particular, Multi-BK-Net achieved a higher mean accuracy of 87.01% on the dedicated test set, which was also statistically significantly higher than that of the baseline CNNs ($p < 0.001$, Mann-Whitney U test). We also observe that the Multi-BK-Net achieved statistically significantly higher mean sensitivity (84.25%) and mean F2-score (85.17%) than each of the baseline CNNs ($p < 0.001$, Mann-Whitney U test). Additional classification results are presented in Appendix A.6.1. For a more in-depth analysis of model performances, we report the classification performances on subsets created based on age and sex information in Section A.6.5, Table 18 and Table 19. With minor exceptions (e.g., for patients younger than 25 years in the TUAB), Multi-BK-Net achieved higher mean accuracies and sensitivities than the baseline models across all subsets. In addition, we report mean ROC curves of the Multi-BK-Net and all five baseline models on the TUAB and TUABEXB test set in Figure 4. The Multi-BK-Net achieved the highest AUC of 0.874 and 0.870 on the TUAB and TUABEXB, respectively, indicating superior sensitivity and overall robustness. ShallowNet and TCN follow closely, whereas Deep4Net achieved the lowest AUC. Figure 5 shows the mean PR curves of the Multi-BK-Net and all five baseline models on the TUAB and TUABEXB test set, respectively. On both test sets, the Multi-BK-Net achieved the highest AUC scores of 0.821 on the TUAB and 0.828 on the TUABEXB. This indicates that Multi-BK-Net achieves superior discriminative performance and reliability compared to the baseline architectures. To evaluate model performance in a clinically meaningful manner, we report sensitivity at a fixed specificity of 95% in Table 2. This metric reflects the true positive rate achieved when the false positive rate is constrained to 5%, corresponding to an acceptable level of false alarms in clinical practice. Unlike the overall AUC, which summarises performance across all thresholds, sensitivity at high specificity provides a threshold-specific measure that aligns with real-world diagnostic requirements and allows for a more relevant comparison between models. Compared with the baseline architectures, Multi-BK-Net achieved the highest mean sensitivities at a fixed specificity of 95% on both test sets (82.78% and 84.25%). The smaller standard deviations further suggest less variability between runs and thus more consistency, improved robustness and accuracy compared to baseline models. In general, our method achieved higher classification performance than five baseline architectures and was more effective in identifying pathological recordings.

## 3.5 Performance Comparision to Previously Reported State-of-the-art Approaches

To further evaluate the Multi-BK-Net, we compared its classification performance with that of other state-of-the-art end-to-end approaches reported in previous studies. More details on the previously published studies are provided in Appendix A.6.2. Table 3 and Table 4 summarise the comparisons between our Multi-BK-Net and other methods on the TUAB and TUABEXB datasets, respectively. Overall, the Multi-BK-Net achieved performance comparable to or better than other deep learning approaches across all three evaluation metrics, with small advantages in some settings. In particular, for the TUAB evaluation set, the following observations can be made by further analysis of the Table 3: First, compared to the four single-scale CNNs, namely Deep4Net, ShallowNet, TCN and EEGNet (Darvishi-Bayazi et al., 2023; Gemein et al., 2020; Khan et al., 2022; Kiessner et al., 2023; 2024; Schirrmeister et al., 2017a; Western et al., 2021), Multi-BK-Net improves the mean classification accuracy by 1.85%, 2.24%, 1.26% and 3.08%, respectively. Second, compared with the three architectures that use three convolutional scales, i.e., ChronoNet (Roy

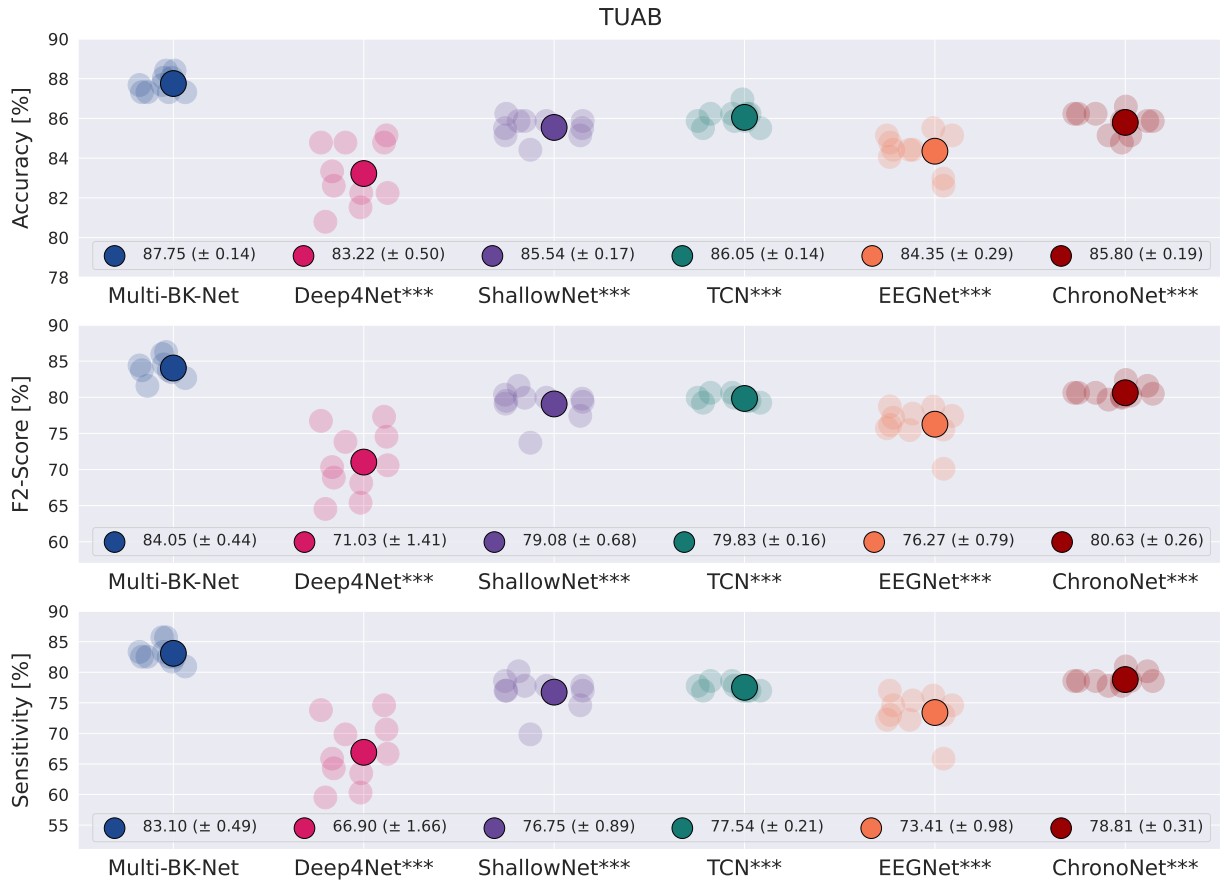

Figure 2: Performance comparison between our proposed Multi-BK-Net and five baseline architectures on the predefined TUAB evaluation set. Each transparent marker represents the performance of a single run, and each larger, bold symbol represents the mean performance score averaged across ten independent runs (n=10). The mean standard error is given in parentheses. Stars indicate statistically significant differences in performance score between the corresponding baseline architecture and the Multi-BK-Net (Bonferroni-corrected two-sided Mann-Whitney U test, $p < 0.05$: *, $p < 0.01$: **, $p < 0.001$: ***; n=10). For additional results, see Figure 8, Appendix A.6.1.

et al., 2019a), XceptionTime model (Brenner et al., 2024), and IRCNN (Wu et al., 2021), Multi-BK-Net improves the classification accuracy by 1.18%, 1.65%, and 0.65%, respectively. In other words, the Multi-BK-Net achieved a classification accuracy of 87.75%, thus outperforming the best previously reported state-of-the-art performance of the IRCNN (Wu et al., 2021). Third, Multi-BK-Net also demonstrates higher classification accuracy than two pre-trained transformer-based foundation models, BIOT (3.2M parameters) and LaBraM (396M parameters), both of which require substantial additional EEG data for pre-training and substantial computational resources due to their large number of parameters. Finally, perhaps the most clinically important finding is that, with a mean sensitivity of 83.10%, Multi-BK-Net outperforms all other approaches, with a mean sensitivity at least 3.26% higher (see Table 3). Furthermore, for the TUABEXB evaluation set, we have the following observations from further analysis of Table 4: Compared with previous results (Kiessner et al., 2023; 2024), the Multi-BK-Net achieved higher mean accuracy (87.01%) and sensitivity (84.25%), representing improvements of at least 1.66% and 2.76%, respectively. Thus, Multi-BK-Net achieved new state-of-the-art results on the predefined TUABEXB evaluation set, setting a new benchmark. Additional results and a more detailed discussion are provided in Appendix A.6.2. Overall, we observe that the Multi-BK-Net outperforms previously reported results on both datasets in terms of mean accuracy. In addition, Multi-BK-Net achieved a higher mean sensitivity of 83–84%, indicating improved

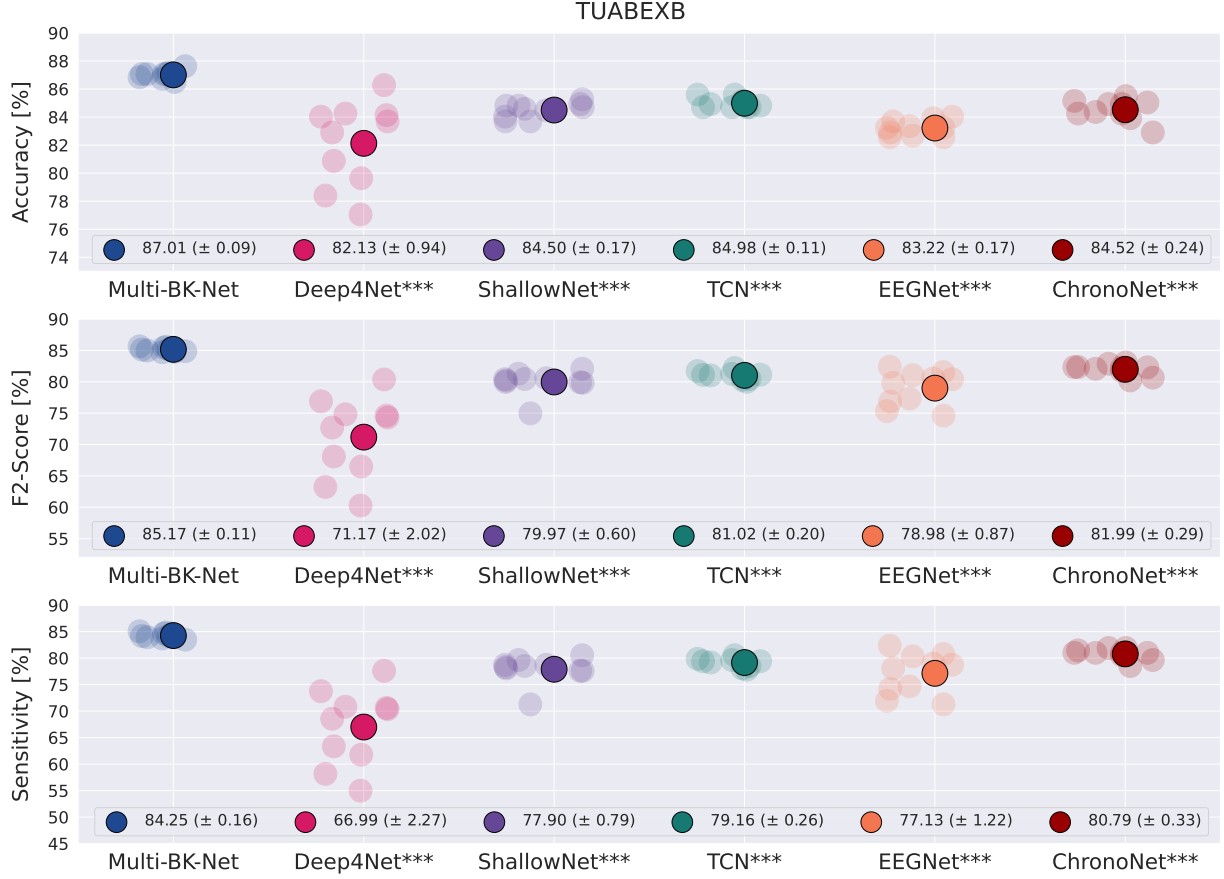

Figure 3: Performance comparison between our proposed Multi-BK-Net and five baseline architectures on the predefined TUABEXB evaluation set. Conventions as in Figure 2. For additional results, see Figure 9, Appendix A.6.1.

performance for real-world medical applications, particularly for medical screening methods where high sensitivity is crucial for accurately identifying the presence of EEG pathology.

## 3.6 Ablation Experiments

To further validate the efficacy of the proposed Multi-BK-Net and to highlight the importance of the multi-scale and multi-branch components in the Multi-BK-Net, we performed several ablation experiments, where we a) compared the performance of the Multi-BK-Net to larger variants of the Deep4Net having the same width or approximately the same size as the Multi-BK-Net ("larger Deep4Net variants"); b) removed four of the five branches of the Multi-BK-Net (Single-Temporal-Kernel Single-Branch Net, STKSBNet, "single temporal kernel + single branch") while changing the number of filters from 7 to 35; and c) used five different temporal kernel lengths but one branch, i.e. concatenating the output of the five temporal convolution layers before the spatial convolution layer (Multi-Temporal-Kernel Single-Branch Net, MTKSBNet, "multiple temporal kernels + single branch"). The results of these experiments can be seen in Table 5 and Table 6, while Appendix A.6.4 provides more details on the ablation experiments, the size of the models (see Table 17) and a more detailed discussion of the results. In comparison to the Multi-BK-Net, "larger Deep4Net variants" achieved statistically significantly lower mean accuracies and mean sensitivities (Table 5), indicating that the superiority of the Multi-BK-Net over the baseline models was not due to the increased model size alone. When compared against our Multi-BK-Net (Table 6), the networks with "single temporal kernel + single branch" and "multiple temporal kernel + single branch" suffered from a statistically significant drop in

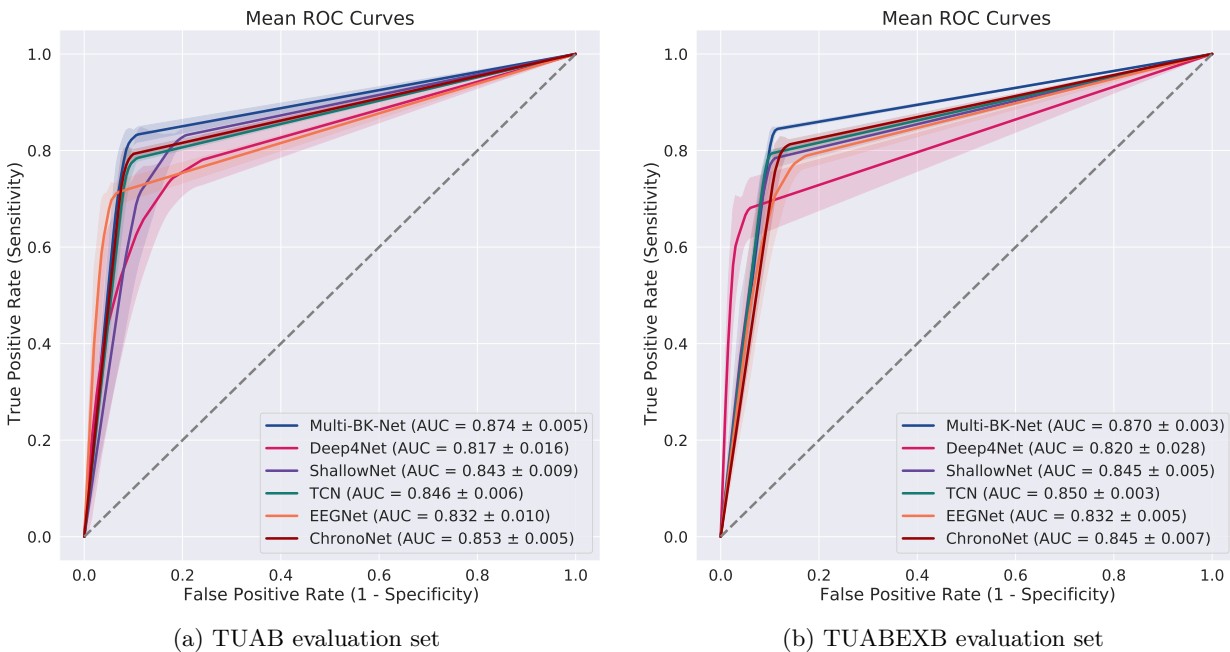

(a) TUAB evaluation set          (b) TUABEXB evaluation set

Figure 4: Mean Receiver Operating Characteristic (ROC) curves of our proposed Multi-BK-Net and five baseline architectures on the predefined test sets of the TUAB (left) and of the TUABEXB (right), averaged across 10 independent runs. Each curve represents the mean interpolated ROC performance across runs (n=10), with shaded areas indicating ±1 standard deviation. The identity line is shown in a grey dotted line. Mean ± SD values of the area under the curve (AUC) are shown in the legend for each model.

classification performance, especially in terms of mean accuracies and mean sensitivities, thereby highlighting the importance of these components towards classification.

### 3.7 Network Interpretation using a UMAP Visualisation

To interpret the proposed network, we extracted the learned features from the last convolution-pooling block of the Multi-BK-Net and visualised them using the Uniform Manifold Approximation and Projection (UMAP) (McInnes et al., 2018) method, a dimensionality reduction technique well-suited to visualising high-dimensional data, such as feature representation vectors. Figure 6 shows the UMAP visualisation of the learned features. As shown in the plot, the representations of the pathological and non-pathological samples form distinct, compact clusters with minimal overlap, indicating that our proposed Multi-BK-Net is highly robust at classifying pathological samples and extracts well-classifiable features, which partly explains its good performance. Further analysis in Appendix A.6.3 reveals that compared to Multi-BK-Net, the three baseline models (Deep4Net, TCN and ChronoNet) tend to form smaller and more dispersed clusters, possibly indicating greater within-class variability or a less precise representation of the pathological class. These results further align with the quantitative performance metrics, such as accuracy and sensitivity, shown in Section 3.4. Overall, the UMAP visualisation of the learned features further supports the efficacy of our method in extracting features and identifying pathological samples.

### 3.8 Network Interpretation using an Amplitude Gradient Analysis

In this section, we investigate how sensitive the Multi-BK-Net's pathological prediction is to amplitude changes in the input. Previous neurophysiological studies and glossaries have described the most common abnormal EEG patterns, for example, amplitude changes within multiple frequency bands (Amin et al., 2023; Emmady & Anilkumar, 2023; Hoppe, 2018; Kane et al., 2017; Medithe & Nelakuditi, 2016; Tatum & William, 2021). Hence, amplitude changes are useful for EEG classification. To determine how sensitive

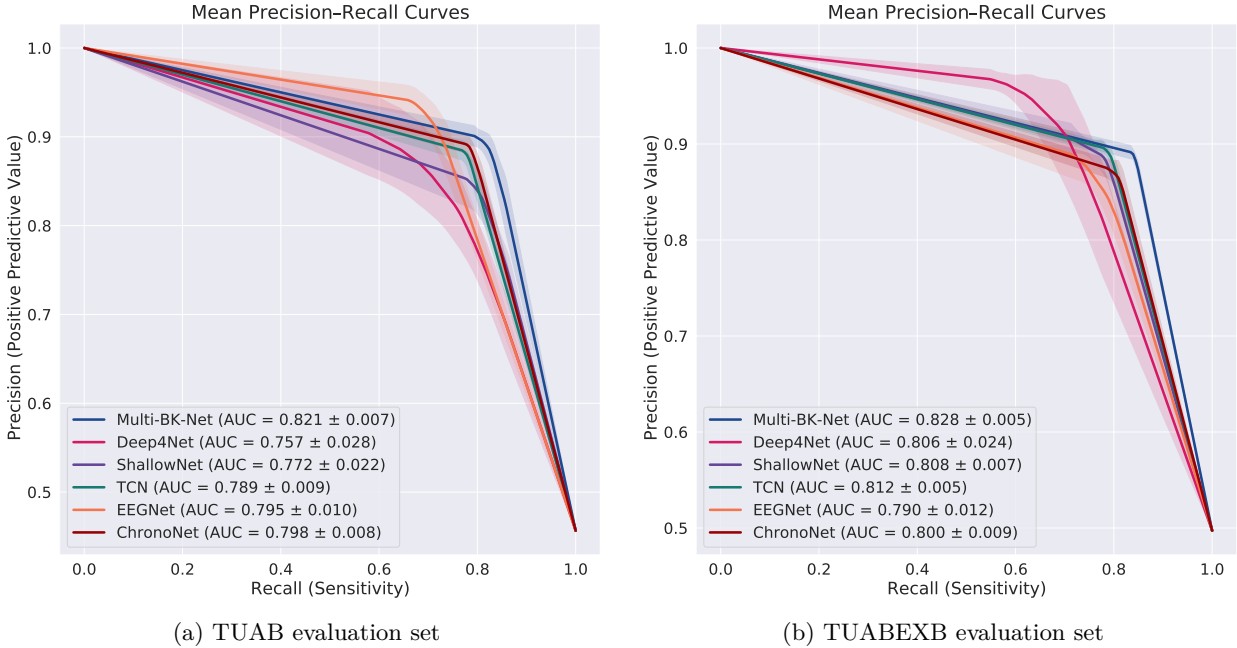

(a) TUAB evaluation set

(b) TUABEXB evaluation set

Figure 5: Mean Precision–Recall (PR) curves of our proposed Multi-BK-Net and five baseline architectures on the predefined test sets of the TUAB (left) and of the TUABEXB (right), averaged across 10 independent runs. Each curve represents the mean interpolated PR performance across runs (n=10), with shaded areas indicating ±1 standard deviation. Mean ± SD values of the area under the PR curve (PR-AUC) are shown in the legend for each model.

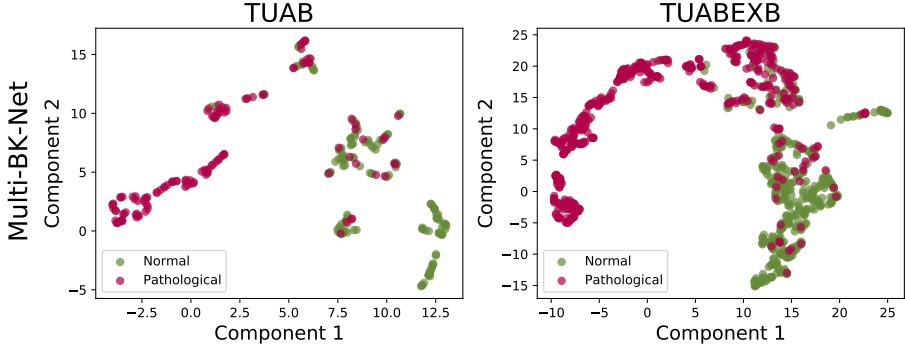

Figure 6: UMAP visualisation of feature representations learned with Multi-BK-Net for pathological and non-pathological recordings on the predefined TUAB (left column) and the TUABEXB (right column) evaluation sets. Pink dots represent the pathological class, while green dots represent the non-pathological class.

the Multi-BK-Net prediction is to amplitude changes across different frequency bands, we calculated the gradients of the output with respect to the input amplitudes, i.e., the recordings from both the TUAB and TUABEXB evaluation sets (Ancona et al., 2018; 2019; Gemein et al., 2020; Schirrmeister et al., 2017a). The gradients were then grouped by pathological status and averaged over all ten runs. The gradients of the model prediction with respect to the input amplitudes linearly approximate how the class prediction responds

Table 3: Performance comparisons of the Multi-BK-Net with previously reported state-of-the-art deep learning methods on the TUAB dataset. Accuracy (ACC), sensitivity (SENS) and specificity (SPEC) scores are given in %, n.a.: not available. Results marked with ⋈ are reported as balanced accuracy. *R*: Publicly available reimplementation. Studies marked with ⚹ used data augmentation. Studies marked with ◁ used only the first minute of each recording for performance evaluation. Studies marked with ♮ used a cropped training strategy as described by Schirrmeister et al. (2017a;b). Studies marked with ∅ reported mean results averaged across several independent runs. Results marked with † are the performance on TUAB using the TUABEXB training set (Kiessner et al., 2023) instead of the TUAB training set. Results marked with ‡ are the performance on TUAB using the TUABCOMB (Kiessner et al., 2024) for training. Results marked with ◇ are averaged across ten runs (n=10).

| Study | Architecture | ACC [%] | SENS[%] | SPEC [%] |
|---|---|---|---|---|
| Schirrmeister et al. (2017a) ♮ ∅ | Deep4Net | 85.40 | 75.12 | 94.13 |
| | ShallowNet | 84.50 | 77.32 | 90.53 |
| Roy et al. (2018) ◁ ∅ | 1D-CNN–RNN | 82.27 | n.a. | n.a. |
| Roy et al. (2019a) ∅ | ChronoNet | 86.57 | n.a. | n.a. |
| Gemein et al. (2020)♮ ∅ | TCN | 86.10 | 79.70 | 91.60 |
| | Deep4Net | 84.60 | 75.90 | 91.90 |
| | ShallowNet | 84.10 | 79.70 | 87.90 |
| | EEGNet | 83.40 | 72.10 | 92.90 |
| Western et al. (2021) ∅ | Deep4Net | 85.90 | 77.00 | 93.30 |
| Wu et al. (2021) ◁ ∅ | IRCNN | 87.10 | n.a. | n.a. |
| Khan et al. (2022) | ChronoNet *R* | 81.00 | n.a. | n.a. |
| | ShallowNet | 85.00 | n.a. | n.a. |
| | Deep4Net | 84.00 | n.a. | n.a. |
| | Hybrid CNN+LSTM | 85.00 | n.a. | n.a. |
| Darvishi-Bayazi et al. (2023) ⚹∅ | Deep4Net | 81.64 ⋈ | n.a. | n.a. |
| | ShallowNet | 82.40 ⋈ | n.a. | n.a. |
| | TCN | 81.69 ⋈ | n.a. | n.a. |
| | EEGNet | 81.40 ⋈ | n.a. | n.a. |
| Kiessner et al. (2023) ♮ ∅ | Deep4Net | 85.51 | 75.95 | 93.53 |
| | ShallowNet | 84.13 | 79.84 | 87.73 |
| | TCN | 85.72 | 78.81 | 91.53 |
| | EEGNet | 84.67 | 72.94 | 94.53 |
| | Deep4Net † | 84.89 | 70.95 | 96.60 |
| | ShallowNet † | 85.51 | 77.30 | 92.40 |
| | TCN † | 86.49 | 77.06 | 94.40 |
| | EEGNet † | 83.51 | 77.22 | 88.80 |
| Yang et al. (2023) ⋈ | BIOT | 79.59 ⋈ | n.a. | n.a. |
| Kiessner et al. (2024) ♮ ∅ | Deep4Net‡ | 85.29 | 73.73 | 95.00 |
| | ShallowNet‡ | 85.22 | 76.83 | 92.27 |
| | TCN‡ | 86.09 | 77.70 | 93.13 |
| | EEGNet‡ | 83.80 | 78.41 | 88.33 |
| Brenner et al. (2024) ◁ | XceptionTime model | 85.10 | 85.70 | 84.70 |
| Jiang et al. (2024) ⋈ | LaBraM-Huge | 82.58 ⋈ | n.a. | n.a. |
| **Proposed method∅◇** | **Multi-BK-Net** | **87.75** | **83.10** | **91.93** |

to changes in the inputs. For example, if the amplitude at a certain channel is changed by a value x, the model output for the class pathological will change approximately by gradient*x. The resulting patterns linearly approximate how changes in input amplitude affect the model's prediction of pathology, indicating which changes in input amplitude are most informative or interesting for the task of classifying general EEG pathology. In addition, we analysed the clinical EEG reports included in the TUAB and TUABEXB datasets.

Table 4: Performance comparisons of the Multi-BK-Net with previously reported state-of-the-art deep learning methods on the TUABEXB dataset. Accuracy (ACC), sensitivity (SENS) and specificity (SPEC) scores are given in %, n.a.: not available. Conventions as in Table 3.

| Study | Architecture | ACC [%] | SENS[%] | SPEC [%] |
|---|---|---|---|---|
| Kiessner et al. (2023) ♮ ∅ | Deep4Net | 83.94 | 71.99 | 95.75 |
| | ShallowNet | 84.47 | 79.32 | 89.55 |
| | TCN | 83.73 | 77.49 | 89.91 |
| | EEGNet | 82.38 | 80.79 | 83.96 |
| Kiessner et al. (2024) ♮ ∅ | Deep4Net ‡ | 85.35 | 75.27 | 95.32 |
| | ShallowNet‡ | 84.58 | 79.79 | 89.31 |
| | TCN‡ | 85.08 | 79.12 | 90.98 |
| | EEGNet‡ | 82.65 | 81.49 | 83.80 |
| **Proposed method∅◇** | **Multi-BK-Net** | **87.01** | **84.25** | **89.73** |

Table 5: Ablation experiment "larger Deep4Net variants". Comparison of the classification performance of Multi-BK-Net with larger Deep4Net variants on the predefined TUAB and TUABEXB evaluation sets. Deep4Net as defined in Schirrmeister et al. (2017a;b). Deep4Net35 (No. start filters: 35); Deep4Net47 (No. start filters: 47); Deep4Net48 (No. start filters: 48). Performance metrics are averaged across ten independent runs (n=10). The mean standard error is given in parentheses. ACC: Accuracy, BACC: Balanced accuracy, F2: F2-score, SENS: Sensitivity, SPEC: Specificity. Stars indicate statistically significant differences in performance between the corresponding model variant and the Multi-BK-Net (Bonferroni-corrected, one-sided Mann-Whitney U test, $p < 0.05$: *, $p < 0.01$: **, $p < 0.001$: ***; n=10).

| Dataset | Architecture | ACC [%] | BACC [%] | F2 [%] | SENS [%] | SPEC [%] |
|---|---|---|---|---|---|---|
| TUAB | Deep4Net | 83.22*** | 81.92*** | 71.03*** | 66.90*** | 96.93 |
| | Deep4Net35 | 84.78*** | 83.63*** | 74.21*** | 70.40*** | 96.87 |
| | Deep4Net47 | 84.89*** | 83.90*** | 75.72*** | 72.46*** | 95.33 |
| | Deep4Net48 | 84.96*** | 83.94*** | 75.52*** | 72.14*** | 95.73 |
| | **Multi-BK-Net** | **87.75** | **87.36** | **84.05** | **83.10** | **91.93** |
| TUABEXB | Deep4Net | 82.13*** | 82.04*** | 71.17*** | 66.99*** | 97.09 |
| | Deep4Net35 | 83.52*** | 83.45*** | 74.59*** | 70.88*** | 96.02 |
| | Deep4Net47 | 84.50*** | 84.44*** | 76.73*** | 73.42*** | 95.46 |
| | Deep4Net48 | 83.87** | 83.80** | 75.14*** | 71.52*** | 96.09 |
| | **Multi-BK-Net** | **87.01** | **86.99** | **85.17** | **84.25** | **89.73** |

Figure 7 shows scalp maps of the amplitude gradients across different frequency bands for the pathological class. The largest absolute values are observed in the delta (0-4 Hz), theta (4-7 Hz) and alpha (8-12 Hz) frequency bands, indicating that the model's pathological prediction is most sensitive to amplitude changes in these specific frequency bands. In particular, in the delta and theta frequency bands, peaks corresponding to positive gradients are observed at the temporal (T3, T4) electrode locations. Specifically, an increase in amplitude in the temporal areas led to an increase in the prediction of the pathological class. This localised pattern is also consistent with clinical EEG reports, which often state that recordings are classified as pathological due to the presence of non-epileptiform abnormalities, such as diffuse or focal slowing in the theta or delta bands (Emmady & Anilkumar, 2023; Nayak & Anilkumar, 2020). Moreover, the clinical EEG reports often cite the "absence of posterior dominant rhythm" as a reason for pathological labelling. Similarly, a negative gradient peak in the alpha band was observed in the occipital brain region, indicating that the predicted probability of the pathological class decreases as the alpha-band amplitude at occipital electrode locations increases, which, in turn, suggests the presence of the PDR. Overall, the results align with current neurophysiological knowledge on pathological EEG patterns, pathological patterns identified by human experts in clinical EEG reports, as well as recent studies that implicate the role of low-frequency

Table 6: Ablation experiments "single temporal kernel + single branch" and "multiple temporal kernels + single branch". Comparison of the classification performance of STKSBNet variants, MTKSBNet variant and Multi-BK-Net on the predefined TUAB and TUABEXB evaluation set. STKSBNet uses a single kernel size (e.g., STKSBNet7: kernel size of 7) and a single branch in the first block. MTKSBNet uses five different kernel sizes, but one branch in the first block. Performance metrics are averaged across ten independent runs (n=10). The mean standard error is given in parentheses. Conventions as in Table 5.

| Dataset | Architecture | ACC [%] | BACC [%] | F2 [%] | SENS [%] | SPEC [%] |
|---------|--------------|---------|----------|--------|----------|----------|
| TUAB | STKSBNet3 | 84.71*** | 84.33*** | 81.00** | 79.92** | 88.73** |
| | STKSBNet7 | 84.13*** | 83.76*** | 80.46** | 79.44** | 88.07*** |
| | STKSBNet10 | 84.96*** | 84.57*** | 81.20** | 80.08* | 89.07** |
| | STKSBNet13 | 86.45** | 85.99** | 82.17* | 80.71** | 91.27 |
| | STKSBNet25 | 86.27** | 85.81** | 82.00** | 80.56** | 91.07 |
| | STKSBNet200 | 86.38*** | 85.92*** | 82.09** | 80.63** | 91.20 |
| | MTKSBNet | 86.59** | 86.16** | 82.55* | 81.19* | 91.13 |
| | **Multi-BK-Net** | **87.75** | **87.36** | **84.05** | **83.10** | **91.93** |
| TUABEXB | STKSBNet3 | 82.73*** | 82.72*** | 81.48*** | 80.93*** | 84.52*** |
| | STKSBNet7 | 84.85*** | 84.83*** | 82.60*** | 81.54*** | 88.12** |
| | STKSBNet10 | 86.09** | 86.07** | 83.76*** | 82.62*** | 89.51 |
| | STKSBNet13 | 85.94** | 85.82** | 83.58** | 82.44** | 89.40 |
| | STKSBNet25 | 85.95*** | 85.93*** | 83.65*** | 82.53*** | 89.33 |
| | STKSBNet200 | 86.16** | 86.14** | 83.62*** | 82.38*** | 89.91 |
| | MTKSBNet | 86.46 | 86.44 | 84.26* | 83.19* | 89.69 |
| | **Multi-BK-Net** | **87.01** | **86.99** | **85.17** | **84.25** | **89.73** |

oscillations in CNNs' classification of general EEG pathology (Gemein et al., 2020; Nahmias & Kontson, 2020; Schirrmeister et al., 2017a).

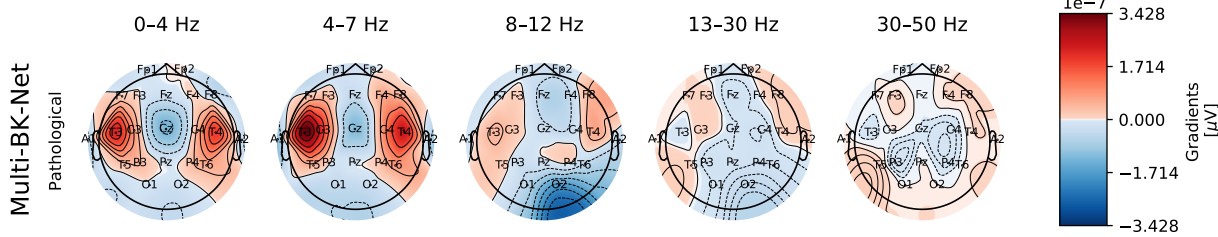

Figure 7: Gradients with respect to the input amplitudes for the pathological class with respect to different frequency bands for the Multi-BK-Net on the combined TUAB and TUABEXB evaluation sets. Delta (0-4 Hz), theta (4-7 Hz), alpha (8-12 Hz), beta (13-30 Hz) and low gamma (30-50 Hz) frequency bands. The red colour indicates positive gradients, while the blue colour indicates negative gradients for the pathological class. Scalp maps are scaled to the absolute maximum value. However, the amplitude gradient results reflect the behaviour of the trained model, and any interpretation of the data itself must be made carefully (Schirrmeister et al., 2017b).

## 3.9 Insights from the Textual EEG Reports

To better understand which EEG recordings are challenging for our Multi-BK-Net to classify, we qualitatively analysed the textual EEG reports of each recording that was misclassified. In particular, for each run of our Multi-BK-Net, we identified all false-positive and false-negative recordings and extracted the corresponding EEG reports. In total, 147 pathological recordings (TUABEXB: 116 recordings; TUAB: 31 recordings) were falsely classified as non-pathological (false negatives, FN), while 139 non-pathological recordings (TU-

ABEXB: 113 recordings; TUAB: 26 recordings) were falsely classified as pathological (false positives, FP). We performed a qualitative analysis of the clinical EEG reports independently for both FN and FP, focusing on the impression and clinical correlation sections.

The qualitative analysis of the FN clinical EEG reports reveals several commonalities due to which the Multi-BK-Net might incorrectly classify them as non-pathological: i) Failure to detect subtle, small, mild abnormalities: Most notably, terms such as "small amount", "very mildly" or "subtle" were used to describe EEG abnormalities that, while not overtly prominent, were still significant enough to be noted clinically. It is possible that this subtlety of change caused the classification errors ("a small degree of background organisation", "this is a very subtle abnormal EEG", "some subtle focal slowing", "this is a mildly abnormal EEG"). ii) Missing single, infrequent or intermittent events: In some cases, single or infrequent events were the reasons why an EEG recording was rated abnormal by the human expert ("A single generalized discharge noted in stage II sleep.", "infrequent, shifting focal slowing is present.", "rare bursts of shifting slowing"). Abnormalities described as "infrequent" or "single" could be missed by CNNs, as the model might average them out or treat them as statistical outliers. It is also possible that these events are not present in the data, as we used only the first 21 minutes of the recordings for training and evaluation (see Section 3.1). iii) Misinterpretation of "nonspecific" abnormalities: Within the FN EEG reports, many abnormalities listed in the impression section are described as "etiologically nonspecific". This implies that even while an abnormality is present, it does not clearly indicate a specific disease. It is therefore possible that the CNN might struggle to classify these less-specific but still abnormal findings as within the range of normal variability, resulting in a false negative prediction. iv) Lack of clinical context: In EEG reports for FN recordings, clinicians include additional information to help interpret EEG patterns as pathological, such as patients' age, current medications, or clinical history. For example, experts often correlate observed EEG patterns with the patient's clinical history. Thus, some nonspecific abnormalities are rated as indicating pathology based on the clinical history of the patient ("The overall record is relatively unremarkable. The 7.8 Hz activity is of unclear significance and may represent a normal variant. Given the past history of neurologic pathology, however, it may also represent evidence of an underlying diffuse disturbance of cerebral function."). In contrast, without a clinical context, the CNN classifies these recordings as normal. Also, the term "age" is consistently used as a factor in determining EEG patterns as pathological, illustrating an age-dependent interpretation ("very mildly abnormal EEG for an adult of this age." "This EEG is just outside of the range of normal for a subject of this age"). This suggests that certain physiological variations are considered pathological in specific age groups. In addition, clinicians often list EEG patterns that could be due to medication ("The minimal excess theta identified above is nonspecific, but could be associated with this patient's medications.","The findings described above could be due to this patient's medications.", "The slowing of the alpha rhythm may be due to the patient's medications."). It is possible that CNN can recognise medication-induced patterns (Nahmias et al., 2020) and thus classify recordings as non-pathological. Moreover, reports often mentioned terms such as "sleep" or "drowsiness", indicating that the experts used the patient's (partial) sleep during the recording to interpret the EEG. During drowsiness, eye movement artefacts, such as blinks and slow or roving eye movements, are commonly observed and can complicate EEG analysis (Mohammedi et al., 2023).

Qualitative analysis of FP clinical EEG reports reveals several commonalities that may cause Multi-BK-Net to misclassify normal recordings as pathological: i) Limitations of routine EEG: The primary reasons listed for "normal" classification are often "the absence of definite epileptiform activity or significant focal abnormalities during routine recording". Frequently, clinical suspicion remains high despite a normal routine EEG, and further comprehensive evaluation is recommended in the clinical EEG reports. The FP reports frequently emphasised the nuances and limitations of EEG interpretations, particularly when dealing with conditions such as epilepsy or seizures. While reports often conclude the findings described as "essential normal" or "within normal limits", they consistently include reservations and recommendations for further clinical correlation ("Midly diffuse cerebral dysfunction cannot be ruled out", " A normal routine EEG does not exclude the diagnosis of epilepsy.", "This is an essentially normal awake and drowsy 1-hour EEG, though a mild diffuse cerebral dysfunction cannot be ruled out."). ii) Misinterpretation of nonspecific findings: Terms like "etiologically nonspecific" mentioned in the EEG reports indicate abnormal findings ("There were no specifically pathological sharp waves observed.","The amplitude is mildly high for the patient's age, but this is nonspecific."). The CNN might detect these abnormal patterns but cannot recognise their nonspecific

nature, leading to a false-positive classification. iii) Lack of clinical context: As noted above in our FN analysis, the EEG reports of the FP also frequently include contextual information about the patient's age, state (awake, drowsy, sleep), medication and clinical history to classify the recording as non-pathological. For example, using information from the patient's clinical history, experts correlate EEG patterns with known skull defects and rate the breach rhythm as a benign variant secondary to structural changes. In contrast, without the contextual information of a pre-existing skull defect, the CNN might classify these patterns as abnormal rather than a benign variant. Also, EEG patterns are rated as benign variants based on the patient's age ("EEG within normal limits for age", "within in normal limits", "EEG at the limits of normal for a subject of this age"). While these are not explicitly listed as reasons for an abnormal EEG, reports often mention "mild diffuse cerebral dysfunction" or "minor shifting slowing in drowsiness", indicating that even subtle non-epileptiform findings have been noted and are frequently correlated with the patient's history ("Occasional mild independent bitemporal slowing is not considered abnormal in the patients above 65 years of age."). While human experts rate abnormalities as clinically insignificant based on additional context, the CNN might identify them as abnormal patterns in the absence of such context. This further supports the assumption that the lack of clinical context is a crucial factor in the CNN's misclassification of EEG recordings. Information about sleep and drowsiness indicates that experts used the fact that the patients were (partially) asleep during the recording to interpret the EEG ("EEG at the limits of normal primarily in drowsiness." "The EEG is remarkable for excess drowsiness. However, when the patient can be awakened, her pattern is within normal limits", "This EEG is compatible with a sedated, sleep pattern."). During drowsiness, physiologically normal but diagnostically challenging changes occur that can closely resemble pathological findings, such as slowed background frequencies and increased slow eye movements, complicating automated or visual EEG analysis (Amin et al., 2023; Strijbis et al., 2022). In contrast, the CNN lacks access to patient information, including the patient's state, age, medications, and clinical history. As a result, the model may misclassify age-appropriate or medication-related variations as pathological abnormalities. iv) Presence of large artefacts: In some FP reports, the presence of large artefacts in the EEG obscuring significant portions of the EEG channels is given as the reason why features are under-recognised or cannot be reliably identified and thus additional studies are recommended ("This EEG include electrode artifact on the right as well as an inability to stimulate the patient adequately. Consequently focal features may be under recognised."). It is possible that the Multi-BK-Net can recognise abnormal EEG patterns despite the presence of large artefacts (Kalita et al., 2024; Tjepkema-Cloostermans et al., 2025).

Overall, this qualitative review suggests that the Multi-BK-Net struggles to identify context-dependent abnormalities. Clearly, our Multi-BK-Net, which was only trained on the EEG signal, does not have access to this contextual information. This puts the model at a disadvantage relative to clinicians, emphasising the potential benefit of incorporating contextual information, such as age and vigilance, into the training and classification processes. Moreover, integrating ratings from multiple experts to train and evaluate the model using averaged ratings, so-called "fuzzy labels", which represent a probability of pathology, might further enhance performance and better reflect diagnostic uncertainty in EEG interpretation (Stephansen et al., 2018). For example, converting classification tasks to probabilistic predictions using fuzzy labels for ambiguous samples (Xu et al., 2024) can help to identify when a model is less specific, guiding clinicians on which samples might require additional manual review (Kang et al., 2021). For a more detailed discussion, see Section 4.

### 3.10 Cross-Institutional Generalisation Performance

Although TUABEXB is larger and more diverse than the TUAB, both datasets originate from the same institution, the Temple University Hospital (TUH), raising concerns about cross-institutional generalisation. At the same time, the possibilities to empirically investigate generalisation performance across multi-institutional datasets remain limited due to the lack of publicly available datasets labelled for general EEG pathology classification. One open-source dataset for general EEG pathology classification that is not derived from the TUH and that contains the same subset of 21 EEG channels as the TUAB and TUABEXB dataset is the NUST-MH-TUKL (NMT) EEG dataset (Khan et al., 2022). The NMT contains 2,417 recordings collected from a South Asian demographic, comprising 2,417 unique patients at the Pak-Emirates Military Hospital. The NMT training set is imbalanced, with 325 pathological and 1907 non-pathological recordings.

Table 7: Cross-institutional generalisation performance comparison between our proposed Multi-BK-Net and five baseline architectures, which were trained on TUH data and applied to the predefined NMT evaluation set. Mean scores are averaged across ten independent runs (n=10). ACC: Accuracy, BACC: Balanced accuracy, F2: F2-score, SENS: Sensitivity, SPEC: Specificity. The standard deviation is given in parentheses. Stars indicate statistically significant differences in performance between the corresponding baseline model and the Multi-BK-Net (Bonferroni-corrected, two-sided Mann-Whitney U test, $p < 0.05$: *, $p < 0.01$: **, $p < 0.001$: ***; n=10).

| Architecture | ACC [%] | BACC [%] | F2 [%] | SENS [%] | SPEC [%] |
|---|---|---|---|---|---|
| Deep4Net | 49.73($\pm$2.25)*** | 50.26($\pm$1.86)*** | 69.6($\pm$28.23) | 82.78($\pm$35.51) | 17.74($\pm$37.54) |
| ShallowNet | 52.02($\pm$6.19)** | 52.62($\pm$5.76)** | 75.85($\pm$16.16) | 89.0($\pm$24.24) | 16.24 ($\pm$34.44) |
| TCN | 54.75($\pm$ 2.84)*** | 54.03($\pm$2.94)*** | 11.86($\pm$10.32)*** | 10.0($\pm$9.04)*** | 98.06($\pm$3.39)*** |
| EEGNet | 50.82($\pm$0.0)*** | 50.0($\pm$0.0)*** | 0.0($\pm$0.0)*** | 0.0($\pm$0.0)*** | 100.0($\pm$0.0)*** |
| ChronoNet | 58.20($\pm$4.41)** | 57.76($\pm$4.61)*** | 33.62($\pm$19.94)*** | 33.62($\pm$19.94)*** | 84.19($\pm$14.35)** |
| **Multi-BK-Net** | **67.32 ($\pm$2.3)** | **67.46 ($\pm$2.17)** | **73.12 ($\pm$3.65)** | **75.89 ($\pm$6.18)** | **59.03 ($\pm$10.28)** |

The NMT evaluation set contains 90 pathological and 95 non-pathological recordings (for more information about the dataset, please refer to Khan et al. (2022)).

To evaluate cross-institutional generalisation performance, we assessed the classification performance of the Multi-BK-Net and all baseline models trained on TUABCOMB on the predefined NMT evaluation set. For more details on performance evaluation, see Section 3.3. We preprocessed the NMT evaluation set data as described in Section 3.1. We observed significant performance degradation across all architectures when applied to the NMT dataset, as shown in Table 7, but less so for our proposed Multi-BK-Net compared to the baseline CNNs. Compared with the five baseline architectures, Multi-BK-Net achieved the highest accuracy, balanced accuracy, and F2-score. Although Deep4Net and ShallowNet demonstrated slightly higher sensitivity, their specificity was considerably lower. Both models achieved an accuracy of around 50%, suggesting they perform at chance level. Overall, all models performed significantly worse on the NMT than on the TUAB and TUABEXB evaluation sets. This also accords with previous studies, which showed that the performance of deep learning approaches trained on the TUAB degrades significantly when they are evaluated on the NMT without any prior exposure to the target dataset and vice versa (Darvishi-Bayazi et al., 2023; Khan et al., 2022). Khan et al. (2022) have evaluated the classification performance of Deep4Net and ShallowNet on NMT after training the models on the TUAB dataset and observed a degradation in performance (48-50% accuracy), indicating a distribution shift. In addition, the authors discussed several factors that may have contributed to this cross-institutional variability, including differences in demographics and the use of other hardware for data collection (for a more detailed discussion, please refer to Khan et al. (2022)). Similarly, Darvishi-Bayazi et al. (2023) observed performance degradation in four CNNs trained on the TUAB when applied to the NMT evaluation set, and vice versa. On the other hand, for training and testing models on the NMT, the accuracies are in the range of 70-80% (Darvishi-Bayazi et al., 2023; Khan et al., 2022). Table 8 summarises the comparison of cross-institutional generalisation performance between our Multi-BK-Net and other methods reported in previous studies. When compared with previously reported results shown in Table 8, the Multi-BK-Net demonstrated superior accuracy of 67.32%, compared with the cross-institutional generalisation performance reported by other studies, which ranged from 45.00% to 63.04%. Together, these results suggest that current deep learning-based approaches are at risk of institutional bias, which limits their generalisability and could lead to performance differences based on demographics, technical, and environmental factors (Bomatter & Gouk, 2025; Saab et al., 2020; Tjepkema-Cloostermans et al., 2025). Furthermore, Khan et al. (2022) demonstrated that fine-tuning can mitigate the impact of cross-institutional variability. Darvishi-Bayazi et al. (2023) also observed that pretraining models on the TUAB and fine-tuning them on NMT enhances the performance on the NMT evaluation set, but at the same time significantly reduces the performance on the TUAB evaluation set, indicating a negative transfer. This shows that there is still room for improvement in transferring knowledge from TUH to NMT. Overall, these results emphasise the need for strategies to mitigate these challenges, as well as for more publicly available datasets from multiple institutions. We discuss this topic further in Section 4.

Table 8: Cross-institutional generalisation performance comparisons of the Multi-BK-Net with state-of-the-art deep learning methods previously reported, which were trained on TUH data and applied to the NMT dataset. Accuracy (ACC), sensitivity (SENS) and specificity (SPEC) scores are given in %, n.a.: not available. Conventions as in Table 3.

| Study | Trained on | Architecture | ACC [%] | SENS[%] | SPEC [%] |
|---|---|---|---|---|---|
| Khan et al. (2022) | TUAB | Deep4Net | 48.00 | n.a. | n.a. |
| | TUAB | ShallowNet | 45.00 | n.a. | n.a. |
| Darvishi-Bayazi et al. (2023) ⋊∅ | TUAB | Deep4Net | 62.77⋈ | n.a. | n.a. |
| | TUAB | ShallowNet | 62.02⋈ | n.a. | n.a. |
| | TUAB | TCN | 63.04⋈ | n.a. | n.a. |
| | TUAB | EEGNet | 61.45⋈ | n.a. | n.a. |
| **Proposed method∅◇** | **TUABCOMB** | **Multi-BK-Net** | **67.32** | **75.89** | **59.03** |

## 4  Discussion and Conclusion

In this work, we demonstrated that through the incorporation of a) multiple temporal kernel lengths based on five clinically relevant frequency bands and b) multiple parallel branches within the first convolution-pooling block, our proposed CNN can outperform five baseline methods in classifying general EEG pathology, while being more effective at capturing the heterogeneity of pathological EEG recordings than their single-scale counterparts. To that end, our Multi-BK-Net extracted task-discriminative long-term and short-term spatiotemporal EEG features for general EEG pathology classification, achieving higher performance than comparable deep end-to-end state-of-the-art methods on two public datasets. The ablation experiments further highlight the efficacy of the multi-branch, multi-scale components in our proposed CNN architecture. Additionally, the UMAP visualisation of the learned features shows that the Multi-BK-Net forms more compact and distinct features than the baseline CNNs, which partly explains the good performance of this architecture. Moreover, the correspondence between the model's sensitivity to localised patterns of amplitude changes in different frequency bands and both current neurophysiological knowledge of pathological EEG patterns and the pathological patterns identified by human experts in the clinical EEG report increased the reliability of the method.

This work addresses a significant gap in CNN approaches that has hindered their development into robust, broadly applicable deep learning-based general EEG pathology classifiers. While CNN-based methods have shown state-of-the-art performance on the task of general EEG pathology classification (Darvishi-Bayazi et al., 2023; Gemein et al., 2020; Khan et al., 2022; Kiessner et al., 2023; 2024; Van Leeuwen et al., 2019; Western et al., 2021; Wu et al., 2021), their classification performance is limited mainly due to the high intra- and inter-subject variability of the EEG signal (Lashgari et al., 2020; Nahmias et al., 2019; Schirrmeister et al., 2017b), the heterogeneity of general EEG pathology patterns (Emmady & Anilkumar, 2023; Nayak & Anilkumar, 2020) and the small size and and limited diversity of the TUAB dataset commonly used for training and evaluating the networks (Kiessner et al., 2023; Poziomska et al., 2025). We demonstrated here that an appropriately constructed CNN, incorporating a multi-scale and multi-branch network design, can achieve more accurate and reliable classification performance than state-of-the-art CNN methods. This is evident from the high mean evaluation accuracies and sensitivities achieved by our method on two public datasets from the Temple University Hospital, as well as from our ablation experiments, thus highlighting its efficacy in classifying general EEG pathology. Our model addresses the challenges inherent to EEG, including its heterogeneity.

An improvement in evaluation sensitivity demonstrates further practical gains of our method. Our proposed Multi-BK-Net achieved mean sensitivities of 83.10% and 84.25% on the TUAB and TUABEXB, respectively, outperforming all other approaches by 3.26% and 2.76%, while attaining high classification accuracies. In comparison to the baseline architectures, Multi-BK-Net achieved the highest mean sensitivities at a fixed specificity of 95% on both test sets (82.78% and 84.25%). The significance of this result lies in applications such as deep learning-based EEG classification and clinical decision support systems in clinical practice, for which high, robust sensitivity is crucial to ensure accurate identification of EEG pathology. In deep learning-

based clinical EEG analysis, high accuracy and sensitivity are of most significant importance because they directly translate into improved patient outcomes, reduced diagnostic errors, and enhanced clinical efficiency. High sensitivity ensures that critical events, such as seizures, are not missed, preventing potential patient harm and delays in treatment (Iešmantas & Alzbutas, 2020; Wang et al., 2023). High accuracy is crucial for reliable diagnosis and proper patient management. Thus, even minor improvements in these metrics are valuable because they can significantly reduce false detections, achieve expert-level agreement, and expedite physician analysis, making the systems more trustworthy and practical for everyday clinical use (Hogan et al., 2025; King-Stephens, 2024; Saab et al., 2020). Such advancements can also lead to better trade-offs between sensitivity and false-positive rates, allowing clinicians to tune the system for optimal performance in specific scenarios (Wang et al., 2023). To be applied in real-world clinical practice, the proposed method must be reliable and utilise robust, well-classifiable features to achieve high performance across a diverse patient population. That is why the evaluation on a larger, more heterogeneous dataset was central in this work. Moreover, we interpreted the model using an amplitude gradient study, which shows that the model's pathological prediction is sensitive to localised patterns of amplitude changes, aligning well with current neurophysiological knowledge of abnormal EEG patterns (Emmady & Anilkumar, 2023; Kane et al., 2017; Nayak & Anilkumar, 2020) and pathological patterns mentioned in clinical EEG reports. In addition, the UMAP visualisation of the learned features further supports the efficacy of our method in extracting features and identifying pathological samples.

While our findings show promise, the broader applicability of these results is subject to certain limitations. For instance, the current study is based on data obtained from a single institution, i.e. Temple University Hospital (TUH). Although using public datasets such as the TUH EEG Corpus (TUEG), which was collected directly in a clinical setting over 14 years and exhibits wide variation in essential parameters such as patient age and diagnosis, is a vital first step in developing deep learning methods that represent the variability of EEG data in real-world applications, it is important to acknowledge several significant limitations of single-institution datasets, such as the potential for institutional bias and demographic performance disparities, that can hinder the effectiveness and real-world applicability of the models. Specifically, models trained solely on data from a single institution may exhibit reduced performance when applied to data from other institutions due to distribution shifts, i.e., differences in patient demographics, acquisition protocols, or equipment (Darvishi-Bayazi et al., 2023; Khan et al., 2022; Poziomska et al., 2025). These discrepancies between training and test data can significantly degrade the model's accuracy and reliability in practical applications (Vishwanath et al., 2023). In addition, limited data diversity in the training data can prevent the model from learning features that are broadly applicable across a wide range of clinical scenarios, thereby restricting its utility to a specific context, institution, or patient population. Thus, models trained on single-institutional data may lack the robustness needed to handle the inherent variability in real-world clinical settings (Li et al., 2025). Consequently, the reported performance might not accurately reflect the model's ability to classify EEG data from new, unseen patients or institutions, thereby overestimating the model's clinical utility (Kamrud et al., 2021; Del Pup et al., 2025; Shafiezadeh et al., 2023). Indeed, these challenges have already been demonstrated in evaluating cross-institutional generalisation performance between datasets from Temple University Hospital and the NMT dataset from Pak-Emirates Military Hospital (Darvishi-Bayazi et al., 2023; Khan et al., 2022, see also Section 3.10). Darvishi-Bayazi et al. (2023) highlighted the challenges associated with negative transfer, while also emphasising key components for overcoming distribution shifts and potential spurious correlations. Building on the insights into cross-dataset transfer learning provided by Darvishi-Bayazi et al. (2023), further studies should explore additional transfer learning strategies. To this end, diverse datasets from multiple institutions and countries are needed to better understand institutional bias and the risks of demographic disparities, and to develop robust and widely applicable models. Also, while our proposed method demonstrates strong performance on a large and diverse public dataset, the absence of validation in real-world clinical settings (e.g., clinician feedback or pilot deployment) is another limitation, thereby constraining conclusions about the model's practical adoption or usability in routine clinical workflows. The issues of cross-institutional validation and practical deployment in real-world clinical settings are intriguing and, therefore, an essential next step to establish the real-world utility and broader applicability of multi-branch, multi-kernel CNNs for general EEG pathology classification.

Despite its limitations, this study certainly contributes to the advancement of deep learning-based general EEG pathology classification. This paper describes our efforts to improve the classification performance

and robustness of CNNs for EEG pathology classification on diverse patient population. To this end, we proposed the Multi-BK-Net, a novel network architecture with multiple temporal kernels and branches in the first block to address the challenges posed by variability and heterogeneity inherent in EEG signals. The enhanced performance, combined with promising capabilities for handling EEG heterogeneity, suggests that the Multi-BK-Net has significant potential to improve the practicality of deep learning-based general EEG pathology classification. Overall, our results demonstrate the efficacy of multi-branch, multi-scale CNN solutions for classifying general EEG pathology by discovering task-relevant features in recorded brain activity, thereby better capturing heterogeneity across pathological samples.

Further performance improvements could be achieved by exploiting advances in the training of CNNs. Some of these include the incorporation of (self-)attention methods (Altaheri et al., 2023a; Liu et al., 2024; Petit et al., 2021) or increasing the size and diversity of the data by using data from multiple sources (Aerts et al., 2017; Poziomska et al., 2025; Schinkel et al., 2023). In addition, the use of medications by most patients poses an additional challenge for deep learning-based EEG classification and should be addressed in further studies (Kamrud et al., 2021). Future research could use Large Language Models to extract more information from clinical EEG reports, yielding more valuable insights, more precise diagnostic explanations, and additional information about the clinical context. Additionally, inspection of the textual reports emphasised the importance of integrating contextual information, such as medication use or clinical history. As human experts also consider patient-specific information (e.g. age, medication, clinical history) during EEG analysis (Beuchat et al., 2021; Kane et al., 2017; Limotai et al., 2020), future research could extend our methods to incorporate contextual information about the patients' age, medication and clinical history, leveraging approaches similar to those explored by Joo et al. (2023), Samak et al. (2023) and Thapa et al. (2024).

Moreover, there is abundant room for further progress in determining whether the use of multi-rater annotations or fuzzy labels can enhance model performance and more accurately reflect diagnostic uncertainty in general EEG pathology classification. Multi-rater annotations can improve performance for several key reasons. First, consensus annotations from multiple experts can serve as a "gold standard" to train and evaluate the performance of deep learning approaches in clinical EEG classification (Hogan et al., 2025) and to provide a more reliable ground truth than single-expert labelling (Stephansen et al., 2018). Second, high-quality, labelled data is essential for developing accurate and reliable deep learning models. The use of labels from multiple expert raters enables the creation of high-quality datasets, which, in turn, allow models to achieve human-expert-level performance across various clinical tasks (Ge et al., 2021). Multiple raters can be used to estimate human-level performance empirically. It can also improve label quality by reducing data noise. A systematic optimisation of labelling quality might even help to overcome the current asymptotic limits of predictive accuracy observed in general EEG pathology classification (Darvishi-Bayazi et al., 2023; Kiessner et al., 2024; Poziomska et al., 2025), and hence represent promising avenues for exploration. Moreover, imperfect or ambiguous annotations from a diverse group of experts can be effectively used to train deep learning models. For example, Saab et al. (2020) have shown that combining various ratings, even if individually imperfect, can improve seizure detection performance. Furthermore, multi-rater annotations reflect diagnostic uncertainty. Annotations from multiple raters can be used to quantify inter-rater agreement (Beuchat et al., 2021), thereby implicitly quantifying the degree of diagnostic uncertainty among human experts, against which deep learning models can be compared. Reflecting diagnostic uncertainty is also essential for clinical adoption. Understanding inter-rater variability provides a benchmark for evaluating the performance of deep learning approaches relative to the range of human expert opinions (Hogan et al., 2025). For example, model performance can be compared with human inter-rater agreement to measure the impact of replacing an expert with deep learning models on overall agreement (Hogan et al., 2025). The potential of deep learning-based approaches for clinical support is evident when the models align with expert ratings (Tjepkema-Cloostermans et al., 2025).

Lastly, with multi-rater annotations, fuzzy labels and fuzzy logic systems can be used for model training and evaluation. These systems provide probabilistic rather than binary classifications of pathology, introducing ambiguity and uncertainty into EEG interpretations. The use of fuzzy labels can improve model robustness and performance, particularly in challenging scenarios. Fuzzy logic systems are well-suited for handling the noise, nonlinearity, and high variability inherent in EEG signals (Ahmadieh et al., 2025; Sorkhi et al., 2022). Also, fuzzy labels can lead to enhanced feature extraction and classification, resulting in superior accuracy,

precision, recall, and F-scores compared to single-rater label models (Gu & Cao, 2020; Shoeibi et al., 2022). For example, fuzzy-logic-based classification algorithms have achieved improved performance and specificity in classifying Alzheimer's disease, Mild Cognitive Impairment, and healthy subjects (Amezquita-Sanchez et al., 2021). Furthermore, fuzzy labels better reflect diagnostic uncertainties. The use of fuzzy logic systems enables the representation of partial truths and degrees of membership in categories (Hu et al., 2021), making them an ideal framework for capturing the inherent ambiguities and variabilities in EEG analysis. In addition, they can be used to quantify uncertainty, which can be incorporated into models to handle it (Ahmadieh et al., 2025). This enables greater tolerance to noisy conditions and addresses uncertainty arising from the high variability of EEG patterns over time (Sorkhi et al., 2022). Recently, there has been increased interest in using uncertainty quantification in deep learning-based methods for medical applications to address diagnostic uncertainty (de Jong et al., 2025). For example, converting classification tasks to probabilistic predictions using fuzzy labels for ambiguous samples (Xu et al., 2024) can help to identify when a model is less specific, guiding clinicians on which samples might require additional manual review (Kang et al., 2021). The fact that deep learning models cannot currently represent predictive uncertainty is a recognised challenge (Prince et al., 2023) that warrants further investigation. In summary, previous studies indicate that the use of multi-rater annotations and fuzzy labels can significantly improve model performance and better reflect diagnostic uncertainty in EEG interpretation. By providing richer, more robust training data, multi-rater annotations can help to improve model performance. In addition, multi-rater annotations are essential for evaluating models against human-level performance and for understanding the inherent variability in human EEG analysis. Overall, multi-rater annotations and fuzzy labels are valuable strategies that help build more robust, expert-aligned models by capturing the full scope of human interpretations and intrinsically modelling the ambiguity and inherent uncertainty in EEG signals, leading to more nuanced predictions and potentially enhanced accuracy, especially in clinically challenging scenarios. Together, these approaches contribute to the creation of more reliable and clinically relevant deep learning-based systems for clinical EEG analysis. A natural progression of this work is therefore to integrate consensus labels or probabilistic (fuzzy) targets, which could mitigate label noise issues, improve model performance, and better reflect diagnostic uncertainty in EEG interpretation.

More broadly, research is also needed to validate deep learning-based EEG classification systems in real-world clinical settings. While large, heterogeneous public datasets such as the TUEG are a valuable source for the initial development and evaluation of deep learning methods for general EEG pathology classification, their translation into impactful clinical (support) systems requires thorough external and real-world validation (El Arab et al., 2025). Pilot studies and validation in real-world clinical settings with clinician feedback are essential before these systems can be reliably applied in everyday clinical practice (El Arab et al., 2025; Hosny et al., 2022; Jacobson & Krupinski, 2021). One way to do this is through pilot studies, which assess the methods' utility, safety, and effectiveness in supporting clinical decision-making and improving patient care. This step ensures practical utility (Zając et al., 2024), clinician usability (McNamara et al., 2024; Olaoye et al., 2024), effectiveness, robustness to EEG variability and other factors (Kelly et al., 2019), and fosters trust (Beger, 2025; Olaoye et al., 2024; Zając et al., 2024). However, deploying deep learning-based methods in clinical practice requires careful consideration of potential risks and challenges. One is the risk of automation bias, where clinicians may rely on model predictions, potentially leading to missed pathology if models are deployed without calibrated thresholds and robust human-in-the-loop guardrails (McLaren et al., 2025; Wang et al., 2020; West et al., 2023). To mitigate these risks, the interpretability of the models is essential, as it enables the explanation of predictions and thereby supports clinical judgement, enhances clinician performance, and is particularly valuable in ambiguous cases (Hasan et al., 2025; Mansilla et al., 2024). Another risk is the institutional bias inherent in models trained on a database from a single institution, which limits their applicability beyond that institutional site and may lead to demographic performance disparities (Bomatter & Gouk, 2025; Darvishi-Bayazi et al., 2023; Khan et al., 2022; Saab et al., 2020; Tjepkema-Cloostermans et al., 2025).

Further studies are needed to elucidate the classification performance and applicability of models across diverse, multi-institutional datasets. Addressing these risks and biases requires data governance and handling protocols. This includes prioritising ethical data sharing, "findable, accessible, interoperable and reusable" (FAIR) principles, and standardised data organisation (Ahluwalia, 2021; Chiang et al., 2021; Ienca et al., 2022; Ochang et al., 2022; Reer et al., 2023). Another clinical challenge is mitigating model drift over

time. For example, changes in clinical practice, data-collection equipment, or demographic changes in the patient population can progressively degrade model performance (Aguilar et al., 2024; Chen et al., 2022; Davis et al., 2022; Gonzalez-Gonzalo et al., 2022; Vela et al., 2022; Vishwanath et al., 2023). Strategies such as evaluating model robustness under realistic distribution shifts, continuous learning, and proactive detection of detrimental data shifts are vital for maintaining the reliability and validity of deep learning-based clinical EEG analysis systems (Ceccon et al., 2024; Feng et al., 2022; Gonzalez-Gonzalo et al., 2022; Guan et al., 2025). Therefore, future clinical implementation must prioritise transparent reporting of model limitations, establish clear protocols of human oversight, and develop adaptive systems that continually learn and remain robust to inherent variability and biases in real-world clinical data (AbuAlrob et al., 2025; Mourid et al., 2025). Nevertheless, real-world application testing is pivotal for validating deep learning methods and is an essential step for translation into everyday clinical practice. Future research should focus on developing methods for seamless clinical integration (Ennab & Mcheick, 2024; Zając et al., 2024), optimising user-interface designs (McNamara et al., 2024; Olaoye et al., 2024; Zając et al., 2024), exploring adaptive learning strategies for continuous improvement against "long-tail" cases (Choudhury & Asan, 2020; Feng et al., 2022) and establishing a comprehensive framework for ongoing performance monitoring and trustworthiness evaluation across diverse real-world settings (Choudhury & Asan, 2020; Feng et al., 2022; Saenz et al., 2024).

## Author Contributions

Ann-Kathrin Kiessner: Software, Writing - original draft, Writing - review & editing, Visualisation, Conceptualisation, Investigation. Tonio Ball: Supervision. Joschka Boedecker: Supervision, Project administration, Resources.

## Acknowledgments

This work is part of BrainLinks-BrainTools which is funded by the Federal Ministry of Economics, Science and Arts of Baden-Württemberg within the sustainability program for projects of the excellence initiative II. Gefördert durch die Deutsche Forschungsgemeinschaft (DFG) - 417962828. Funded by the Deutsche Forschungsgemeinschaft (DFG, German Research Foundation) – 417962828. We acknowledge funding by the Deutsche Forschungsgemeinschaft (DFG, German Research Foundation) under SFB 1597 (SmallData), grant number 499552394 and under AI-Cog-BA 4695/4-1. This work was funded by BMBF Grants: KIDELIR-16SV8864 and DiaQNOS-13N16460.

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

# A   Appendix

## A.1   Related Work

In this section, we briefly summarise the related work on deep learning methods for general EEG pathology classification that have been successfully applied to the TUH Abnormal EEG Corpus (TUAB) (López de Diego, 2017) or TUH Abnormal Expansion Balanced EEG Corpus (TUABEXB) (Kiessner et al., 2023) (see Table 3 and Table 4 in Section 3.5). In recent years, substantial research has explored various approaches to general EEG pathology classification, reporting accuracies within a narrow range of 79% to 87%. However, the majority of previous work is based on the TUAB (see Table 3 in Section 3.5). In this regard, Schirrmeister et al. (2017a) introduced and applied two CNNs, Deep4Net and ShallowNet, originally proposed for motor imagery and adapted for EEG pathology classification, achieving accuracies of 85.40% and 84.50%, respectively. These models[4] have been successfully reused or reimplemeted in other studies (Darvishi-Bayazi et al., 2023; Gemein et al., 2020; Khan et al., 2022; Kiessner et al., 2023; 2024; Van Leeuwen et al., 2019; Western et al., 2021; Wu et al., 2021). Since then, several different architectures have been used to classify general EEG pathology. For instance, Gemein et al. (2020) employed two additional networks: a compact CNN called EEGNet and a temporal convolutional neural network (TCN). The EEGNet has been proposed by (Lawhern et al., 2018) for EEG-based BCIs using depthwise and separable convolutions. The TCN was introduced by (Bai et al., 2018) for sequence modelling and has since been optimised for EEG classification via a neural architecture search (Chrabąszcz, 2018; Gemein et al., 2020). Other studies have employed different CNNs, such as 1-D CNNs (Shukla et al., 2021; Yıldırım et al., 2020), or hybrid models that combine CNN and RNN (Roy et al., 2018; 2019a), an Inception-Residual CNN (Wu et al., 2021), or an XceptionTime model (Brenner et al., 2024). Recently, a few approaches have explored the potential of a transformer-based foundation model for EEG pathology classification. The Large Brain Model (LaBraM) model (Jiang et al., 2024) (369M parameters) and the Biosignal Transformer (BIOT) model (Yang et al., 2023) (3.2M parameters) were pre-trained on a combination of different datasets using a cross-dataset training and evaluated on the TUAB dataset. Moreover, Roy et al. (2019a) proposed a deep 1D convolutional gated recurrent neural network, called ChronoNet, with exponentially varying filter sizes in the Conv1D layers. While ChronoNet achieved one of the highest accuracies (86.57%), its open-source re-implementation achieved a lower performance (81%) (Khan et al., 2022). Similarly, despite reusing the original implementation of some of these CNNs, studies reported slightly different results from the original report. The reasons for these differences include variations in EEG preprocessing, such as other input lengths, different versions of the TUAB, or different datasets, as well as differences in hardware and training strategies. On the TUAB, Wu et al. (2021) reported the highest accuracy of 87.10% using the IRCNN model. Overall, the reported approaches achieved a mean specificity of above 84% and a mean sensitivity of about 75-80% on the TUAB. Most models show a lower sensitivity than specificity, indicating that they are more likely to classify pathological examples as non-pathological than vice versa. Baseline results on the TUABEXB dataset have been reported using Deep4Net, ShallowNet, TCN and EEGNet (see Table 4 in Section 3.5). These four CNNs achieved typical EEG pathology classification accuracies on TUABEXB, with slightly lower accuracies ranging from approx-

---

[4]are available in Braindecode, a deep learning toolbox for EEG, which is available for download at `https://github.com/TNTLFreiburg/braindecode`.

imately 82% to 86%. Consistent with previous studies on the TUAB, the models mostly achieved a lower sensitivity (72-82%) than specificity (83-96%) on the TUABEXB.

Although multi-branch, multi-scale, or parallel architectures (Altuwaijri et al., 2022; Belwafi et al., 2017; Ingolfsson et al., 2020; Jia et al., 2021; Riyad et al., 2020; Szegedy et al., 2015; Zhang et al., 2023a) are now beginning to gain traction in various EEG classification tasks, including motor imagery (Altuwaijri et al., 2022; Cai et al., 2024; Jia et al., 2021; Liu & Yang, 2021; Liu et al., 2023; Yang et al., 2021), emotion recognition (Emsawas et al., 2022; Yan et al., 2025), and neonatal sleep staging (Siddiqa et al., 2024; Zhu et al., 2023), only a few attempts have been made to assess the efficacy of using CNNs with a set of three convolution scales for general EEG pathology classification. For example, Roy et al. (2019a) has considered the importance of convolutional layers with multiple filters of exponentially varying sizes (2, 4, and 8) to extract and combine features from different time scales. Inspired by the core idea of inception (Szegedy et al., 2015; Zhang et al., 2023a), Wu et al. (2021) employed stacked convolution kernels of different, yet small, scales (1, 3, and 5) in parallel. Recently, Brenner et al. (2024) employed an XceptionTime model, in which each module applies a set of three depthwise separable convolutions with different kernel sizes (11, 21 and 41). These studies have reported performance improvements, with accuracies ranging from 85.10% to 87.10%. However, these results were based on the TUAB, and no attempts have been made to evaluate the performance of multi-scale CNNs on a larger, more heterogeneous dataset. In addition, smaller kernel sizes were preferred due to lower computational costs (Emsawas et al., 2022), which, however, tend to extract shorter temporal patterns from faster frequency bands (Cohen, 2014; Jia et al., 2021). In contrast, larger kernel sizes require higher computational costs, while they can learn long-term temporal patterns from slow frequency bands (Cohen, 2014; Jia et al., 2021). To date, research has not yet determined whether using raw EEG signals as input for a multi-branch CNN with multi-scale convolutions, which combine both long-term and short-term temporal patterns by employing kernel sizes based on clinically relevant frequency bands, is effective for classifying general EEG pathology on larger, heterogeneous datasets.

## A.2 Additional Details about the Datasets

In this section, we provide additional details on the TUH Abnormal EEG Corpus (TUAB)[5] (López de Diego, 2017) and the TUH Abnormal Expansion Balanced EEG Corpus (TUABEXB)[6] (Kiessner et al., 2023) that we used in our experiments. The two datasets are publicly available subsets of the Temple University Hospital EEG Corpus (TUEG)[7] (Obeid & Picone, 2016) for general EEG pathology classification, consisting of EEG recordings labelled as non-pathological or pathological. The TUEG comprises 69,582 clinical EEG recordings from 26,873 EEG sessions involving 15,001 patients over 15 years. Details on the number of TUAB recordings and patients are shown in Table 9, while Table 10 gives details of the number of recordings and patients in the TUABEXB dataset. For training, we concatenated the TUAB and TUABEXB training sets, which we refer to as the TUH Abnormal Combined EEG Corpus (TUABCOMB). We evaluated our models on the predefined test sets of the TUAB and TUABEXB datasets, respectively (see Section 3.3). Details on the number of recordings and patients of the TUABCOMB are shown in Table 11. There is no overlap of patients between the training set of TUABCOMB and the evaluation sets of TUAB and TUABEXB.

## A.3 Design Choices and Hyperparameter Optimisation

For our proposed Multi-BK-Net architecture described in Section 2.1, we evaluated several design choices, including architecture hyperparameters, such as total number of temporal filters, filter length of the later convolution layer, strides and types of non-linearities, as well as algorithm hyperparameters, such as learning rate, weight decay and number of training epochs. In addition, we evaluated potential performance improvements by using intermediate normalisation by batch normalisation (Ioffe & Szegedy, 2015) or group normalisation (Wu & He, 2018) as well as the use of exponential linear units (ELU, $f(x) = x$ for $x > 0$ and $f(x) = e^x - 1$ for $x \leq 1$ ) (Clevert et al., 2016) or GELU ($GELU(x) = xP(X \leq x) = x\Phi(x)$) (Hendrycks &

---

[5]v2.0.0; available for download at `https://www.isip.piconepress.com/projects/tuh_eeg/html/downloads.shtml`.

[6]The EEG recordings are part of the TUH EEG Corpus and are publicly available after registration on the TUH EEG Corpus website (`isip.piconepress.com/projects/tuh_eeg/html/downloads.shtml`. The corresponding pathology labels are available for download at `github.com/AKiessner/TUHAbnormal-Expansion-dataset`.

[7]v1.1.0 and v1.2.0; available for download at `https://isip.piconepress.com/projects/tuh_eeg/downloads/tuh_eeg/`.

Table 9: Number of recordings and patients in the TUAB dataset. For 54 patients in the training set, both non-pathological and pathological recordings are available. However, there is no overlap between patients in the training and evaluation sets. For more details on the dataset, see López de Diego (2017) and Obeid & Picone (2016).

| TUH Abnormal EEG Corpus (TUAB) | Training set | | Evaluation set | |
|---|---|---|---|---|
| | Recordings | Patients | Recordings | Patients |
| Non-pathological | 1371 | 1237 | 150 | 148 |
| Pathological | 1,346 | 893 | 126 | 105 |
| Total | 2,717 | 2,130 | 276 | 253 |

Table 10: Number of recordings and patients in the TUABEXB dataset. There are 183 patients with both pathological and non-pathological recordings in the training set. The training and evaluation sets do not share recordings from the same patient. Details of the dataset and the labelling procedure can be found in Kiessner et al. (2023).

| TUH Abnormal Expansion Balanced EEG Corpus (TUABEXB) | Training set | | Evaluation set | |
|---|---|---|---|---|
| | Recordings | Patients | Recordings | Patients |
| Non-pathological | 4,015 | 3,253 | 447 | 392 |
| Pathological | 3,975 | 3,166 | 442 | 378 |
| Total | 7,990 | 6,419 | 889 | 770 |

Gimpel, 2023) as activation function. The hyperparameters used and their possible values, which construct the search space, are listed in Table 12.

To identify the optimal set of the CNN hyperparameters (Table 12), we employed the automatic hyperparameter optimisation framework Optuna (Akiba et al., 2019), as it has been successfully used to efficiently tune the hyperparameters of CNNs (Al-Ja'afreh et al., 2023; Hanifi et al., 2024; Latreche et al., 2025; Shekhar et al., 2021). Optuna implements Sequential Model-Based Optimisation (SMBO) to efficiently search for the optimal hyperparameters by building a probabilistic model of the objective function (Akiba et al., 2019; Bergstra et al., 2011; Lemos et al., 2021). To search for the hyperparameter space and maximise the objective function, we used the Tree-structured Parzen Estimator (TPE) algorithm (Bergstra et al., 2011; 2013). We used multivariate TPE rather than independent TPE because multivariate TPE is reported to outperform independent TPE by finding better solutions faster and by better handling problems where there is an interaction between variables. The optimisation approach is described in detail in Bergstra et al. (2011; 2013); Falkner et al. (2018) and Kenny et al. (2024). In medical diagnosis, identifying all potentially pathological EEG recordings is more important than reducing false-positive results. Therefore, two objective measures — validation accuracy and validation sensitivity — were considered in this study. The multi-objective function for hyperparameter optimisation is defined as the maximisation of the mean validation accuracy and mean validation sensitivity using 5-fold cross-validation. For each set of hyperparameters, we performed 5-fold cross-validation on the TUABCOMB training data using StratifiedGroupKFold from the Scikit-learn library (Pedregosa et al., 2011), with the constraint that patients were non-overlapping across splits (80% for training and 20% for validation). The splits were made at the patient level to avoid using recordings from the same patient in different folds. We additionally shuffled the data before splitting to ensure that splits were not chronologically ordered, similar to the predefined training and evaluation sets in the TUAB and TUABEXB datasets. Experiments were conducted with a time budget of 45 hours per fold for each configuration run on a single fold. Runs that exceeded the time limit were pruned after completing the first fold. Runs that crashed (e.g., network configurations that did not fit in GPU memory) were also pruned. In addition, runs with a validation accuracy or sensitivity below 75% in the first fold were pruned to speed up optimisation. We ran the optimisation process until 100 trials were completed, jointly optimising parameters and architecture layouts. For the top 10 trials, we repeated the 5-fold cross-validation ten times and computed the mean validation accuracy and mean validation sensitivity, averaging across folds and then across runs. Finally, the

Table 11: Number of recordings and patients in the TUABCOMB training set dataset. There are 237 patients with both pathological and non-pathological recordings. The training set of the TUABCOMB and the evaluation sets of the TUAB and TUABEXB do not share the same patients.

| TUABCOMB | Training set | |
|---|---|---|
| | Recordings | Patients |
| Non-pathological | 5,386 | 4,490 |
| Pathological | 5,321 | 4,009 |
| Total | 10,707 | 8,549 |

Table 12: Configuration spaces considered in the search for the Multi-BK-Net architecture. We sampled until 100 configuration trials were completed to find the best setting. Learning rate and weight decay were defined as trial.suggest_float(log=True).

| Parameter | Config. Space |
|---|---|
| Total number of temporal conv filters | [20,25,30,35,40,45,50,55,60,65,70,75,80] |
| Normalisation | [batch, group] |
| Activation functions | [ELU, GELU] |
| Pooling mode first block | [mean, max] |
| Pooling mode other blocks | [mean, max] |
| Forth conv-pooling-block | [True, False] |
| Forth conv-pooling-block broader | [True, False] |
| Dropout | [0.4 - 0.6] |
| Filter length conv blocks | [10,15,20] |
| Input window size | 6000 |
| Weighted loss factor pathological | [1,2,3,4] |
| Optimiser | [AdamW] |
| Optimiser beta1 | [0.5,0.9] |
| Learning rate | $[1e^{-5} - 1e^{-1}]$ |
| Weight decay | $[1e^{-5} - 1e^{-1}]$ |
| Batch size | [16,32,64] |
| Number of epochs | [30 - 105] |
| Number of channels | 21 |

best model configuration, with the highest mean accuracy and mean sensitivity, was used to train the model on the full TUABCOMB training set and then evaluated on the unseen, predefined evaluation sets from the TUAB and TUABEXB (see Section 3.3 for more details). The final hyperparameters are listed in Table 1 in Section 2.1.

### A.4 Additional Details on the Baseline Architectures

In this section, we provide additional details on the baseline architectures, the list of hyperparameters used for training, and the training procedure. We additionally included five architectures implemented in Braindecode as a baseline: Deep4Net (Schirrmeister et al., 2017b;a), ShallowNet (Schirrmeister et al., 2017b;a), TCN (Bai et al., 2018; Chrabąszcz, 2018; Gemein et al., 2020), EEGNet Gemein et al. (2020); Lawhern et al. (2018) and ChronoNet Roy et al. (2019a). These architectures have been utilised in several studies, demonstrating high performance on the TUAB and TUABEXB datasets across various versions and different preprocessing steps. The architectures are shown in Table 14 and Table 15. For a more detailed explanation of each architecture and its optimised hyperparameters, please refer to the corresponding studies. To ensure fairness in evaluation, we apply the same data preprocessing steps and training strategy to all architectures. While we have employed a trial-wise training strategy in this work, previous studies have mainly used a cropped training strategy (Gemein et al., 2020; Kiessner et al., 2023; 2024; Schirrmeister et al., 2017a). To adapt the

Table 13: Optimised hyperparameters of the four single-scale CNNs that we used as a baseline (Gemein et al., 2020; Schirrmeister et al., 2017a; Kiessner et al., 2023; Chrabąszcz, 2018). No: Number of. For Deep4Net, ShallowNet and EEGNet, the parameter final_conv_length was set to 'auto' to enable trial-wise training. For TCN, an AdaptiveAvgPool1d layer was added to the final block.

| Hyperparameter | Deep4Net | ShallowNet | TCN | EEGNet |
|---|---|---|---|---|
| No. input channels | 21 | 21 | 21 | 21 |
| Input time length | 6000 | 6000 | 6000 | 6000 |
| No. start filters | 25 | 40 | n.a. | n.a. |
| No. filters | n.a. | n.a. | 55 | n.a. |
| F1 (temporal filter) | n.a. | n.a. | n.a. | 8 |
| D depth multiplier | n.a. | n.a. | n.a. | 2 |
| F2 (pointwise filter) | n.a. | n.a. | n.a. | 16 |
| Final conv. length | 67 | 394 | n.a. | 187 |
| Channel factor | 2 | n.a. | n.a. | n.a. |
| Stride before pool | True | n.a. | n.a. | n.a. |
| Initial learning rate | 0.01 | 0.000625 | 0.0011261049710243193 | 0.001 |
| Weight decay | 0.0005 | 0 | 5.83730537673086e-07 | 0 |
| Dropout | 0.5 | 0.5 | 0.05270154233150525 | 0.25 |
| L2 decay | n.a. | n.a. | 1.7491630095065614e-08 | n.a. |
| Gradient clip | n.a. | n.a. | 0.25 | n.a. |
| Batch size | 64 | 64 | 64 | 64 |
| No. epochs | 35 | 35 | 35 | 35 |
| **Total No. parameters** | 303,452 | 66,242 | 456,502 | 7,426 |

architectures to our framework, we used their trial-wise training approach (Schirrmeister et al., 2017b). All models were trained on the TUABCOMB training set for 35 epochs using the trial-wise training strategy (Schirrmeister et al., 2017b). For all models, a batch size of 64 was used. The model parameters of Deep4Net, ShallowNet, TCN, and EEGNet were optimised using the AdamW optimiser (Loshchilov & Hutter, 2017). For more details on the hyperparameters used for training, see Table 13. For optimisation of the ChronoNet, we use the Adam optimiser (Kingma & Ba, 2017) with a learning rate of 0.001 and the binary cross-entropy loss. The learning rates for the gradient and weight decay updates were scheduled using cosine annealing (Loshchilov & Hutter, 2016), and we refrained from learning rate restarts.

### A.5   Additional Details on the Evaluation Metrics

In this section, we provide additional details on the evaluation metrics and significance tests used in this study. In our experiments, we evaluated the classification performance of our proposed Multi-BK-Net and comparison models using five performance evaluation metrics: accuracy, balanced accuracy, sensitivity, specificity and F2-score. These measures were determined as shown in Table 16. As both evaluation sets are slightly imbalanced (see Table 9 and Table 10), we also reported the balanced accuracy (BACC). Since our task is to classify EEG pathology, and thus no EEG recording containing pathological EEG patterns should be classified as non-pathological, it is crucial to reduce the false-negative rate. False positives, on the other hand, can be excluded by clinicians through a more detailed diagnostic process. As minimising false negatives is the primary concern in this task, we additionally report the F2-score, which prioritises the model's ability to identify pathological EEG recordings (recall) over the model's ability to identify positive instances while minimising false positives.

Table 14: Architecture of the four single-scale, baseline CNNs adapted for trial-wise training and as implemented in Braindecode (Gemein et al., 2020; Schirrmeister et al., 2017a; Kiessner et al., 2023; Chrabąszcz, 2018). E: Number of input electrodes. St: Stride.

| Deep4Net | ShallowNet | TCN | EEGNet |
|---|---|---|---|
| 25×Conv2D (10×1) | 40×Conv2D (25×1) | 55×Conv1D (16) | 8×Conv2D (1×64) |
| 25×Conv2D (1×$E$) | 40×Conv2D (1×$E$) | WeightNorm | BatchNorm |
| BatchNorm | BatchNorm | Activation (ReLU) | 16×Conv2D ($E$×1) |
| Activation (ELU) | Activation (Square) | Dropout2d (0.25) | BatchNorm |
| MaxPool (3×1) | MeanPool (75×1) St(15×1) | 55×Conv1D (16) | Activation (ELU) |
| | Activation (Log) | WeightNorm | MeanPool (1×4) St(1×4) |
| | Dropout (0.5) | Activation (ReLU) | Dropout (0.25) |
| | | Dropout2d (0.25) | |
| | | 55×Conv1D (1) | |
| | | Activation (ReLU) | |
| Dropout (0.5) | 2×Conv2D (394×1) | 55×Conv1D (16) | 16×Conv2D (1×16) |
| 50×Conv2D (10×1) St(3×1) | LogSoftmax | WeightNorm | 16×Conv2D (1×1) |
| BatchNorm | | Activation (ReLU) | BatchNorm |
| Activation (ELU) | | Dropout2d (0.25) | Activation (ELU) |
| MaxPool (3×1) | | 55×Conv1D (16) | MeanPool (1×8) St(1×8) |
| | | WeightNorm | Dropout (0.25) |
| | | Activation (ReLU) | |
| | | Dropout2d (0.25) | |
| Dropout (0.5) | | 55×Conv1D (55×16) | 2×Conv2D (1×187) |
| 100×Conv2D (10×1) St(3×1) | | WeightNorm | LogSoftmax |
| BatchNorm | | Activation (ReLU) | |
| Activation (ELU) | | Dropout2d (0.25) | |
| MaxPool (3× 1) | | 55×Conv1D (16) | |
| | | WeightNorm | |
| | | Activation (ReLU) | |
| | | Dropout2d (0.25) | |
| Dropout (0.5) | | 55×Conv1D (16) | |
| 200×Conv2D (10×1) St(3×1) | | WeightNorm | |
| BatchNorm | | Activation (ReLU) | |
| Activation (ELU) | | Dropout2d (0.25) | |
| MaxPool (3×1) | | 55×Conv1D (16) | |
| | | WeightNorm | |
| | | Activation (ReLU) | |
| | | Dropout2d (0.25) | |
| 2×Conv2D (67×1) | | 55×Conv1D (16) | |
| LogSoftmax | | WeightNorm | |
| | | Activation (ReLU) | |
| | | Dropout2d (0.25) | |
| | | 55×Conv1D (16) | |
| | | WeightNorm | |
| | | Activation (ReLU) | |
| | | Dropout2d (0.25) | |
| | | Linear (55,2) | |
| | | AdaptiveAvgPool1d | |
| | | LogSoftmax | |

Table 15: Architecture of the baseline architecture ChronoNet as proposed by Roy et al. (2019a). ChronoNet includes multiple filters of exponentially varying lengths in the 1D convolutional layers and dense connections within the GRU layers. Arrows represent skip connections. K: kernel_size. st: stride. p: padding. h: hidden_size

| ChronoNet | | |
|---|---|---|
| 32×Conv1D(k(2,), st(2,),p(1,)) | 32×Conv1D(k(4,), st(2,)) | 32×Conv1D(k(8,), st(2,),p(3,)) |
| | Filter Concat | |
| 32×Conv1D(k(2,), st(2,),p(1,)) | 32×Conv1D(k(4,), st(2,)) | 32×Conv1D(k(8,), st(2,),p(3,)) |
| | Filter Concat | |
| 32×Conv1D(k(2,), st(2,),p(1,)) | 32×Conv1D(k(4,), st(2,)) | 32×Conv1D(k(8,), st(2,),p(3,)) |
| | Filter Concat | |
| | GRU(h(32)) | |
| | GRU(h(32)) | |
| | Filter Concat | |
| | GRU(h(32)) | |
| | Filter Concat | |
| | GRU(h(32)) | |
| | Linear layer | |
| **Total No. parameters** | 137,233 | |

Table 16: Performance evaluation metrics. *TP* is the number of examples that were correctly classified in the positive class, *TN* is the number of examples that were correctly classified in the negative class, *FP* is the number of examples that were incorrectly classified in the positive class, *FN* is the number of examples that were incorrectly classified in the negative class. F2-score: $\beta = 2$ to emphasise the need to minimise false negatives, and $Precision = \frac{\text{TP}}{\text{TP+FP}}$ and $Recall = \frac{\text{TP}}{\text{TP+FN}}$. Higher values of the F2-score indicate better recall.

| Performance metric | Mathematical formula |
|---|---|
| Accuracy | $\dfrac{(\text{TP+TN})}{(\text{TP+FN+TN+FP})}$ |
| Balanced accuracy | $\dfrac{1}{2}\left(\dfrac{\text{TP}}{\text{TP+FN}} + \dfrac{\text{TN}}{\text{TN+FP}}\right)$ |
| Sensitivity | $\dfrac{\text{TP}}{\text{TP+FN}}$ |
| Specificity | $\dfrac{\text{TN}}{\text{TN+FP}}$ |
| F2-score | $(1+\beta^2)\left(\dfrac{\frac{\text{TP}}{\text{TP+FP}} * \frac{\text{TP}}{\text{TP+FN}}}{(\beta^2 * \frac{\text{TP}}{\text{TP+FP}}) + \frac{\text{TP}}{\text{TP+FN}}}\right)$ |

## A.6 Detailed Results

### A.6.1 Performance Comparison to Baseline Approaches

The classification performance of the Multi-BK-Net and all baseline models on the TUAB is shown in Figure 8. On the TUAB evaluation set, Multi-BK-Net achieved a mean balance accuracy of 87.36%, indicating

the best classification performance, which was also statistically significantly better than the baseline models ($p < 0.001$, Mann-Whitney U test). Compared to four of the baseline architectures, our Multi-BK-Net achieved a slightly lower specificity, which was not statistically significantly different, except for the difference with Deep4Net ($p < 0.001$, Mann-Whitney U test). Figure 9 compares the balanced accuracy and sensitivity of our proposed Multi-BK-Net and all baseline architectures on the TUABEXB evaluation set. Again, the Multi-BK-Net outperformed the baseline models. In particular, the Multi-BK-Net achieved a higher mean balanced accuracy of 86.99% on the dedicated test set, which was also statistically significantly different from the baseline architectures ($p < 0.001$, Mann-Whitney U test). Although three of the baseline models achieved a slightly higher specificity than Multi-BK-Net (89.73%), only the difference between Multi-BK-Net and Deep4Net was statistically significant ($p < 0.01$). Conversely, Multi-BK-Net achieved a statistically significantly higher mean specificity than ChronoNet ($p < 0.05$, Mann-Whitney U test).

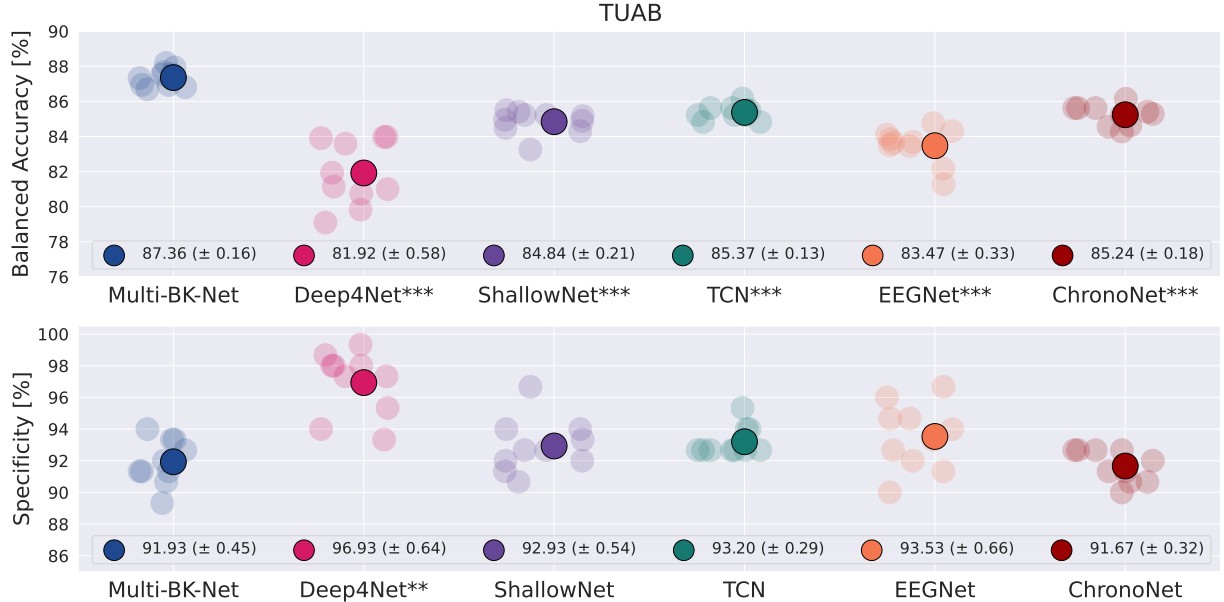

Figure 8: Performance comparison between our proposed Multi-BK-Net and five baseline architectures on the predefined TUAB evaluation set. Each transparent marker represents the performance of a single run, and each larger, bold symbol represents the mean performance score averaged across ten independent runs (n=10). The mean standard error is given in parentheses. Stars indicate statistically significant differences in performance score between the corresponding baseline architecture and the Multi-BK-Net (Bonferroni-corrected two-sided Mann-Whitney U test, $p < 0.05$: *, $p < 0.01$: **, $p < 0.001$: ***).

### A.6.2 Performance Comparison with Previous Reported State-of-the-art Deep Learning Approaches

Several studies have employed various end-to-end deep learning methods for general EEG pathology classification (see Section A.1). In this section, we provide more details on the comparison of our proposed methods to previous work. For the selection of a previously published work on end-to-end deep learning models for general EEG pathology classification, we considered different architectures that have achieved competitive classification performance and match the following criteria: To ensure a fair comparison, the selected models adopt the raw or minimally preprocessed input from the same set of multiple standard scalp EEG electrodes and are evaluated on the predefined TUAB evaluation set or on the predefined TUABEXB evaluation set. Hence, publications that reported cross-validation results (Muhammad et al., 2021; Nahmias & Kontson, 2020; Poziomska et al., 2025), used single-channel data as input (Shukla et al., 2021; Yıldırım et al., 2020), or evaluated their methods on data from a non-publicly available dataset (Kim et al., 2023; Van Leeuwen et al., 2019), were excluded from our direct comparison. Table 3 and Table 4 in Section 3.5 summarise the comparisons between our Multi-BK-Net and other methods on the TUAB and TUABEXB datasets, respectively. Perhaps the most clinically relevant finding is that Multi-BK-Net, with a mean sensi-

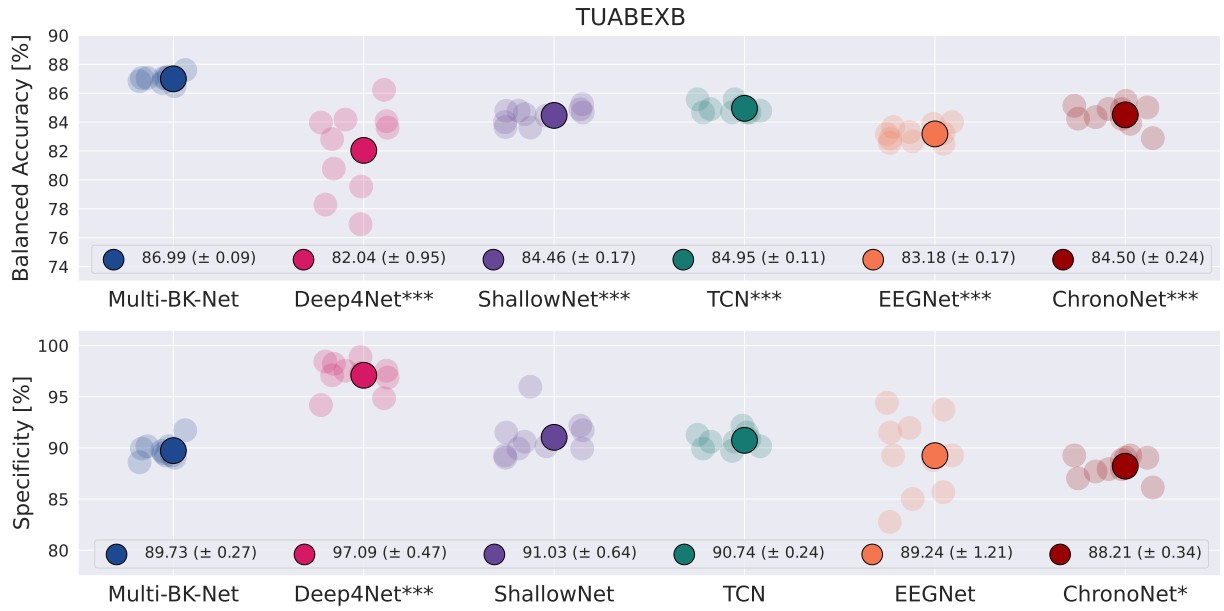

Figure 9: Performance comparison between our proposed Multi-BK-Net and all baseline architectures on the predefined TUABEXB evaluation set. Performance metrics are averaged across ten independent runs (n=10). The mean standard error is given in parentheses. Conventions as in Figure 8.

tivity of 83.10%, outperforms all other approaches, reporting a mean sensitivity averaged over multiple runs, by at least 3.26% (see Table 3). Note that Brenner et al. (2024) reported a sensitivity of 85.70% for the XceptionTime model. However, the authors used only the first window of each recording as input during evaluation and reported single-run results, which is a significant limitation of this approach. Due to the use of stochastic gradient-based algorithms with randomised weight initialisation, data ordering, and data augmentation during training, each independent training run produces a different network with better or worse performance than the average, with a variance of approximately 1% to 2% (Picard, 2023). Even if the variance between runs is not very large, there is a high probability that the performance of a single run will be an outlier, much better or much worse than the average (Picard, 2023). As a result, this variance can make it challenging to compare training configurations (Bouthillier et al., 2021; Picard, 2023). For example, comparing single runs, Multi-BK-Net achieved a sensitivity of 85.72% with accuracies of 88.41% and 87.68%, and a specificity of 90.67% and 89.33%, respectively, thus clearly outperforming the XceptionTime model. Nevertheless, to reduce stochasticity and improve comparability between training and model configurations, one should report the average of evaluation metrics over multiple runs (Wightman et al., 2021). Hence, a direct comparison of publications reporting single-run results would be unfair. Furthermore, comparing Table 3 with Table 4, it can also be seen that the mean accuracy of the Multi-BK-Net is slightly better on the TUAB than on the TUABEXB (+0.74%). At the same time, Multi-BK-Net achieved a higher mean sensitivity on TUABEXB than on TUAB (+1.15%). Interestingly, this result is consistent with previous observations (Kiessner et al., 2023), assuming that the differences are due to systematic differences in age, sex and pathological feature distribution as well as different labelling methods (see Kiessner et al., 2023, for a more detailed discussion). Furthermore, we observed that Multi-BK-Net achieved lower sensitivity than specificity on both evaluation sets, indicating a lower false-positive rate than false-negative rate. This is also consistent with the results presented in previous literature (Gemein et al., 2020; Kiessner et al., 2023; 2024; López de Diego, 2017; Schirrmeister et al., 2017a; Western et al., 2021).

Overall, one of the main findings of our study is that the proposed model achieved mean accuracies of 87–88%. However, as noted and discussed in previous work (Engemann et al., 2018; Frénay & Verleysen, 2014; Gemein et al., 2020; Kiessner et al., 2024; Poziomska et al., 2025; Sun et al., 2019), there remains the possibility that this accuracy range may be the upper limit due to imperfect inter-rater agreement and the resulting

label noise, with a model performance remaining below a perfect classification score (100%) and saturating around 88%. For classifying EEG recordings as pathological or non-pathological, previous research has reported inter-rater agreement of 86–88% between two neurologists (Houfek & Ellingson, 1959; Rose et al., 1973). At the same time, Beuchat et al. (2021) have found even lower mean inter-rater agreements of 82-86% between multiple EEG technologists and neurologists. Unfortunately, the differences in classification performance cannot be compared with the inter-rater agreement on these datasets used in this study for training and evaluation, as the labels of the TUAB and TUABEXB are mainly based on the impression of a single expert, as stated in the corresponding EEG report (for more details, see Kiessner et al., 2023; López de Diego, 2017; Obeid & Picone, 2016). Also, other studies on general EEG classification suffer from the same uncertainty. However, the need to include ratings from multiple experts has been recognised in other EEG classification tasks. For instance, Stephansen et al. (2018) compared the classification performance of their model with the inter-rater agreement among six raters for classifying sleep stages. The authors found that the model's accuracy increased from 76% (one rater) to 87% (consensus of six raters). Therefore, it can be assumed that the deep learning models for general EEG pathology also perform better than a single rater. It would thus be interesting to compare the classification performance of the proposed model with the inter-rater agreement of multiple raters. As the label noise remains the main limiting factor, it will be important to determine whether and to what extent label quality can be improved. For instance, using labels from an ensemble of multiple raters would be a significant step toward improving label quality. Additionally, using averaged ratings (so-called "fuzzy labels") for training could further enhance model performance, as they represent a probability of pathology, which is also helpful in identifying recordings that are ambiguous or more challenging to categorise or contain more confounding patterns. Furthermore, as shown in Beuchat et al. (2021), multiple raters tend to achieve higher mean sensitivity (87%) than mean specificity (75%), the opposite of machine learning approaches, which typically achieve higher specificity than sensitivity. Thus, including multiple raters as a baseline for training could increase the sensitivity of the classification model and also might further improve model performance, as was done for sleep stage classification. Moreover, a systematic optimisation of labelling quality might help the models overcome the current asymptotic limits of predictive accuracy (Darvishi-Bayazi et al., 2023; Kiessner et al., 2024; Poziomska et al., 2025), and hence lead to even better-performing models for general EEG pathology classification.

### A.6.3 UMAP Visualisation of Learned Features

To further explain the proposed network, we visualised the learned features of the Multi-BK-Net before the classification layer using the UMAP (McInnes et al., 2018) method to map high-dimensional feature vectors into a two-dimensional UMAP space. This allowed the visual inspection of clustering and separation between pathological and non-pathological classes. For comparison, we also extracted and visualised the feature representations of the Deep4Net, as it is a single-scale CNN that is most similar in architecture to Multi-BK-Net, the TCN, which achieved the second-highest classification accuracy, and the ChronoNet, which uses a set of three convolutions. A UMAP comparison plot of the visualisation of the feature representation of the last layer before the classification layer is shown in Figure 10. In general, all models can form discernible clusters for both pathological and non-pathological recordings, but with noticeable overlap. This is reflected in a reduced classification performance, which falls within a narrow range of 83% and 88%. As Multi-BK-Net achieves the best classification performance, it also forms distinct, compact clusters for pathological and non-pathological recordings with minimal overlap. Smaller intra-class distances, i.e., tighter clustering, indicate a more consistent representation within each class, while larger inter-class distances, i.e., more distinct clusters, indicate that the model has learned features that effectively discriminate between the two classes. Compared to Multi-BK-Net, the three baseline models tend to form smaller and more dispersed clusters, possibly indicating greater within-class variability or a less precise representation of the pathological class. For example, Deep4Net forms less compact clusters, resulting in more dispersed clusters with larger intra-class distances. In addition, we can observe that pathological clusters are more dispersed along the component 1 axis, which may indicate greater within-class variability for pathological samples. This further suggests that these baseline models are less consistent in identifying pathological samples and less capable of capturing their heterogeneity. The slightly lower capability to capture the characteristics of pathological samples may also reflect the three baseline models' lower sensitivity (higher false-negative rate) compared to Multi-BK-Net. Additionally, when comparing the datasets (left vs. right column, Figure 6), we observe

that all four models exhibit more distinct feature separation and better-defined clusters on the TUAB dataset than on the TUABEXB dataset. This suggests that the learned features are more discriminative for the TUAB dataset than the TUABEXB dataset. For all models, the distinction between pathological and non-pathological recordings was less pronounced on TUABEXB. Since better feature separation leads to improved classification performance, this may result in better model performance on TUAB than TUABEXB. A possible explanation for this could be that the TUABEXB is more heterogeneous compared to the TUAB (Kiessner et al., 2023), and as shown in previous work, this data heterogeneity can make EEG classification more difficult (Kiessner et al., 2023; Poziomska et al., 2025). Finally, the qualitative observations derived from the UMAP visualisations, which highlight the different degrees of feature separation achieved by various models, are consistent with quantitative performance metrics, such as accuracy and sensitivity. For instance, the distinct, compact clustering of data points, indicative of strong feature discrimination, observed for the Multi-BK-Net corresponds to higher accuracy and sensitivity scores than those of other models on both datasets. Conversely, the large intra-class distances observed in the visualisations of the pathological samples of the three baseline CNNs align with their lower sensitivity (67-81%). Similarly, models with poorer cluster separation in the UMAP visualisations, such as the Deep4Net, show comparable poorer quantitative performance (accuracy of 82-83%). This consistency between qualitative visualisations and quantitative results supports the validity of the UMAP analysis and provides a comprehensive understanding of model performance.

### A.6.4 Ablation Experiments

In this section, we provide additional details on the ablation experiments (Section 3.6) and present additional results. Compared to the five baseline architectures, our proposed Multi-BK-Net is significantly larger (1,038,683 parameters). For example, it is approximately twice the size of TCN (456,502 parameters), three times the size of Deep4Net (303,452 parameters), and eight times the size of ChronoNet (137,233 parameters). The Multi-BK-Net achieved an accuracy improvement of 2-5% over the baseline models, while sensitivity increased by more than 3.5-17.3%. Previous studies have shown that larger models outperform smaller ones (Darvishi-Bayazi et al., 2023; Howard et al., 2017; Kiessner et al., 2024; Talmor & Berant, 2019; Tan & Le, 2019; Sun et al., 2017; Zagoruyko & Komodakis, 2016). To verify that the superiority of the Multi-BK-Net over the baseline models was not due to its increased model size, we conducted an ablation study. As the Deep4Net has the most similar architecture to our proposed Multi-BK-Net, we compared the performance of our approach with that of three larger versions of the Deep4Net (see Table 17). In particular, we increased the number of temporal filters in the Deep4Net to 35, the same number used in the first convolution-pooling block of the Multi-BK-Net. We refer to this model as Deep4Net35, which comprises 579,182 parameters, approximately 55% of the size of Multi-BK-Net. We have also increased the number of temporal filters in the Deep4Net to 47 (Deep4Net47). This increases the number of trainable parameters to 1,026,482, approximately 99% of the size of the Multi-BK-Net. Finally, we created a variant called Deep4Net48, which contains 48 temporal filters and has 1,069,490 trainable parameters; thus, it is slightly larger than Multi-BK-Net.

To further validate the efficacy of our multi-temporal kernel branches in the first convolution-pooling block of our proposed Multi-BK-Net, we also trained and evaluated variants of our model in which we removed four of the five branches and increased the number of filters in the remaining branch from 7 to 35. Thus, the variant models consist of a single branch with 35 filters and a single temporal kernel length in the first block. The kernel length of the variants is set to either 3, 7, 10, 13, 25 or 200. The Multi-BK-Net uses kernel lengths of 3, 7, 13, 25 and 200, while the Deep4Net uses a kernel size of 10. We refer to these variants as STKSBNet (Single-Temporal-Kernel Single-Branch Net). For example, STKSBNet3 denotes the model variant with a temporal kernel length of 3. Furthermore, to investigate the effect of different branches, we included an additional variant with five temporal kernels (3, 7, 13, 25, and 200), but used only one branch instead of five. Thus, the features of the five temporal convolutional layers are concatenated directly before the spatial convolution. We refer to this model variant as MTKSBNet (Multi-Temporal-Kernel Single-Branch Net). For more details on the size of the models used, see Table 17. We trained the STKSBNet and MTKSBNet variants as described in Section 2.1, and the Deep4Net variants as described in Section 3.2 and evaluated all variants as described in Section 3.3.

Table 17: Size of the different architectures used in the ablation experiments. Model size is given as the number of trainable parameters.

| Architecture | Model size |
|---|---|
| Deep4Net | 303,452 |
| Deep4Net35 | 579,182 |
| Deep4Net47 | 1,026,482 |
| Deep4Net48 | 1,069,490 |
| STKSBNet3 | 1,057,632 |
| STKSBNet7 | 1,057,772 |
| STKSBNet10 | 1,057,877 |
| STKSBNet13 | 1,057,982 |
| STKSBNet25 | 1,058,402 |
| STKSBNet200 | 1,064,527 |
| MTKSBNet | 1,038,683 |
| **Multi-BK-Net** | **1,038,683** |

As shown in Table 5 in Section 3.6, the performance of Deep4Net increases as the number of filters increases. However, only Deep4Net47 obtained a statistically significantly higher mean F2-score and mean sensitivity on the TUABEXB set compared to Deep4Net ($p < 0.05$, one-sided Mann-Whitney U test). All other differences between the original Deep4Net and its larger variants were not statistically significant ($p > 0.05$, Mann-Whitney U test). Nevertheless, Multi-BK-Net outperforms all variants of Deep4Net statistically significantly on both datasets ($p < 0.01$, Mann-Whitney U test). This suggests that the performance improvement of Multi-BK-Net over Deep4Net is not due to increased model width or size.

Moreover, Table 6 in Section 3.6 shows the performance of the variant models and the Multi-BK-Net on the TUAB and TUABEXB evaluation sets. It can be seen that when four branches and four kernel sizes are removed, there is a significant decrease in the results on both evaluation sets ($p < 0.05$, Mann-Whitney U test). Using a single kernel length, STKSBNets cannot extract accurate features, resulting in lower overall performance than Multi-BK-Net. This highlights the efficacy of using multiple temporal kernel lengths. Furthermore, when comparing the performance of the STKSBNet variants, it is evident that the optimal kernel size varies across datasets. For TUAB, the mean accuracy is highest at a kernel size of 13, whereas the optimal kernel size for TUABEXB is 200. This observation is consistent with previous studies, which report that determining the optimal kernel length for EEG pathology classification is challenging due to subject and time differences (Jia et al., 2021). A possible solution is to use multiple kernels of different sizes, which also improves classification performance (Altuwaijri et al., 2022; Jia et al., 2021). Furthermore, the results show that the inclusion of multiple temporal kernels of different lengths, but with a single branch, improves performance compared to using a single kernel of a single length. This shows that various kernels can learn valuable features and improve classification performance. However, as shown in Table 6, incorporating multiple branches within the first block further improves performance on both datasets compared to using only a single branch (MTKSBNet). When the four branches are removed but multiple kernel lengths are used (MTKSBNet), the mean F2-score and mean sensitivity decrease ($p < 0.05$, Mann-Whitney U test). This shows that introducing multiple input branches improves the identification of pathological samples. Overall, this ablation experiment demonstrates that incorporating multiple temporal kernel lengths and multiple branches, as in Multi-BK-Net, is beneficial for classifying EEG pathology and significantly enhances the model's performance.

### A.6.5 Overview of Classification Performance in Different Age and Sex Subgroups

For a more in-depth analysis of model performance, we report classification performance on subsets defined by age and sex in Table 18 and Table 19. The tables present a detailed performance comparison of the proposed Multi-BK-Net against five baseline architectures across various subgroups of both the TUABEXB and TUAB datasets. Subgroup analyses were conducted by stratifying the test data by age and sex and

Table 18: Performance comparison between our proposed Multi-BK-Net and five baseline architectures on different subgroups of the predefined TUAB evaluation set. Performance metrics were computed separately for each subgroup and averaged across 10 independent runs (n=10). Confidence intervals (95% CI) represent the mean based on the normal approximation (n=10). ACC: Accuracy [%], SENS: Sensitivity [%], SPEC.: Specificity [%]. Bonferroni-corrected, two-sided Mann-Whitney U test, $p < 0.05$: *, $p < 0.01$: **, $p < 0.001$: ***; n=10).

| Group | Architecture | ACC. [95%CI] | SENS. [95%CI] | SPEC. [95%CI] |
|---|---|---|---|---|
| age < 25 y. (n=18) | Deep4Net | 90.56 [88.23-92.88] | 74.00 [65.63-82.37] | 96.92 [94.46-99.39] |
| | ShallowNet | 94.44 [92.15-96.74] | 90.00 [83.47-96.53] | 96.15 [92.78-99.53] |
| | TCN | 93.33 [91.16-95.51] | 94.00 [88.01-99.99] | 93.08 [91.57-94.58] |
| | EEGNet | 92.31 [90.72-93.90] | 76.00 [70.77-81.23] | 96.19 [93.86-98.52] |
| | ChronoNet | 93.89 [91.35-96.43] | 88.00 [79.33-96.67] | 96.15 [93.64-98.67] |
| | Multi-BK-Net | 92.78 [89.51-96.04] | 86.00 [77.63-94.37] | 95.38 [92.05-98.72] |
| 25 y. ≤ age < 60 y. (n=174) | Deep4Net | 83.39***[82.20-84.58] | 69.12** [64.71-73.52] | 92.55 [88.71-96.38] |
| | ShallowNet | 85.86***[85.15-86.58] | 75.44 [73.42-77.46] | 92.55*[90.35-94.74] |
| | TCN | 86.32***[85.88-86.76] | 71.76***[71.05-72.48] | 95.66 [95.17-96.15] |
| | EEGNet | 85.00***[84.48-85.52] | 65.29***[63.73-66.86] | 97.64 [96.95-98.33] |
| | ChronoNet | 88.10 [87.63-88.58] | 73.82** [72.79-74.86] | 97.26 [96.83-97.70] |
| | Multi-BK-Net | 89.08 [88.43-89.73] | 77.65 [76.30-78.99] | 96.42 [95.51-97.32] |
| 60 y. ≤ age (n=84) | Deep4Net | 78.69***[77.24-80.14] | 79.43 [73.16-85.71] | 77.42 [68.38-86.46] |
| | ShallowNet | 79.64** [78.42-80.87] | 88.30 [86.67-89.94] | 64.84* [59.64-70.04] |
| | TCN | 81.19** [80.43-81.95] | 84.34**[83.55-85.13] | 75.81 [74.11-77.51] |
| | EEGNet | 80.95** [80.10-81.80] | 76.60***[74.19-79.02] | 88.39***[85.53-91.25] |
| | ChronoNet | 79.64** [78.52-80.77] | 84.53** [83.32-85.74] | 71.29 [68.55-74.03] |
| | Multi-BK-Net | 83.93 [83.06-84.80] | 89.06 [87.73-90.38] | 75.16 [72.84-77.48] |
| female (n=148) | Deep4Net | 80.74***[79.64-81.85] | 72.86* [66.42-79.29] | 86.59 [81.43-91.75] |
| | ShallowNet | 82.03***[81.12-82.94] | 80.48 [78.06-82.89] | 83.18**[80.26-86.09] |
| | TCN | 84.53***[84.07-84.99] | 76.83***[76.00-77.66] | 90.24 [89.54-90.93] |
| | EEGNet | 84.05***[83.34-84.77] | 69.52***[67.46-71.59] | 94.82**[93.78-95.87] |
| | ChronoNet | 84.66***[84.07-85.26] | 76.83***[75.87-77.78] | 90.47 [89.60-91.34] |
| | Multi-BK-Net | 87.43 [86.94-87.92] | 82.86 [81.33-84.38] | 90.82 [89.50-92.15] |
| male (n=128) | Deep4Net | 84.38***[82.81-85.94] | 74.44** [70.13-78.76] | 94.00 [89.61-98.39] |
| | ShallowNet | 87.42 [86.58-88.26] | 82.38 [80.96-83.81] | 92.31 [89.57-95.04] |
| | TCN | 86.02** [85.35-86.68] | 79.05** [78.14-79.95] | 92.77 [91.76-93.78] |
| | EEGNet | 84.61***[83.85-85.37] | 71.43***[69.41-73.45] | 97.38 [96.19-98.58] |
| | ChronoNet | 87.34 [86.59-88.09] | 80.95 [79.82-82.09] | 93.54**[92.46-94.62] |
| | Multi-BK-Net | 88.12 [87.58-88.67] | 82.70 [81.06-84.33] | 93.38 [92.60-94.17] |

computing the corresponding metrics and confidence intervals for each subgroup. Performance metrics were computed separately for each subgroup and averaged across ten independent runs. Mean performance metrics and mean 95% confidence intervals were then calculated across ten independent runs using the normal approximation (mean $\pm$ 1.96 $\times$ standard error). With minor exceptions (e.g., for patients younger than 25 years at TUAB), Multi-BK-Net achieved higher mean accuracies and sensitivities than the baseline models across all subsets. For instance, in Table 19, the Multi-BK-Net achieves a mean accuracy of 87.51% for the 25 years ≤ age < 60 subgroup and a mean accuracy of 86.82% for the male subgroup. Similarly, in Table 18, the Multi-BK-Net obtain a mean accuracy of 88.43% for the 25 years ≤ age < 60 subgroup and a mean accuracy of 88.12% for the male subgroup. While other architectures, such as TCN and EEGNet, occasionally show very high specificity, they often achieve significantly lower sensitivity. The Multi-BK-Net, in contrast, maintains strong performance across both sensitivity and specificity, indicating its robust ability to identify both pathological and non-pathological recordings across different subgroups correctly.

Table 19: Performance comparison between our proposed Multi-BK-Net and five baseline architectures on different subgroups of the predefined TUABEXB evaluation set. Performance metrics were computed separately for each subgroup and averaged across 10 independent runs (n=10). Confidence intervals (95% CI) represent the mean based on the normal approximation (n=10). ACC: Accuracy [%], SENS: Sensitivity [%], SPEC.: Specificity [%]. Bonferroni-corrected, two-sided Mann-Whitney U test, $p < 0.05$: *, $p < 0.01$: **, $p < 0.001$: ***; n=10).

| Group | Architecture | ACC. [95%CI] | SENS. [95%CI] | SPEC. [95%CI] |
|---|---|---|---|---|
| age < 25 y. | Deep4Net | 81.94** [80.60-83.29] | 49.58***[44.76-54.41] | 98.13** [96.92-99.33] |
| (n=144) | ShallowNet | 81.11** [80.12-82.10] | 66.67** [64.39-68.94] | 88.33 [86.08-90.59] |
| | TCN | 82.78* [81.71-83.85] | 68.12** [66.63-69.62] | 90.10 [88.40-91.81] |
| | EEGNet | 77.88***[76.71-79.06] | 65.56** [62.73-68.38] | 84.41** [81.58-87.24] |
| | ChronoNet | 76.67***[75.50-77.84] | 67.50** [65.65-69.35] | 81.25***[79.10-83.40] |
| | Multi-BK-Net | 85.62 [84.44-86.81] | 72.29 [70.92-73.66] | 92.29 [90.54-94.05] |
| 25 y. ≤ age < 60 y. | Deep4Net | 82.10***[80.37-83.83] | 59.64***[54.93- 64.35] | 98.65***[98.06-99.23] |
| (n=462) | ShallowNet | 84.78***[84.24-85.32] | 69.64***[67.68-71.61] | 95.94***[95.24-96.64] |
| | TCN | 85.45** [84.93-85.98] | 72.55***[71.59-73.51] | 94.96***[94.65-95.28] |
| | EEGNet | 84.22***[83.65-84.79] | 69.74***[66.01-73.48] | 94.89 [92.92-96.86] |
| | ChronoNet | 86.45** [86.12-86.78] | 75.46***[74.34-76.58] | 94.55** [93.92-95.18] |
| | Multi-BK-Net | 87.51 [87.20-87.82] | 80.00 [79.24-80.76] | 93.05 [92.55-93.54] |
| 60 y. ≤ age | Deep4Net | 82.20***[79.68-84.71] | 78.48***[74.15-82.82] | 90.95***[88.51-93.39] |
| (n=282) | ShallowNet | 85.71 [85.35-86.07] | 88.79** [87.57-90.00] | 78.45 [76.13-80.77] |
| | TCN | 85.28** [84.92-85.65] | 88.38***[87.89-88.87] | 77.98 [76.62-79.33] |
| | EEGNet | 84.26** [83.51-85.00] | 86.97***[85.64-88.30] | 77.86 [73.89-81.83] |
| | ChronoNet | 85.32* [84.56-86.08] | 89.29** [88.64-89.95] | 75.95 [74.09-77.82] |
| | Multi-BK-Net | 86.84 [86.19-87.49] | 91.36 [90.99-91.74] | 76.19 [74.19-78.19] |
| female | Deep4Net | 81.11***[79.01-83.21] | 65.96***[61.21-70.70 ] | 97.14***[96.21-98.07] |
| (n=467) | ShallowNet | 83.79***[83.37-84.21] | 76.21***[74.51-77.91] | 91.81* [90.55-93.06] |
| | TCN | 84.58** [84.17-84.99] | 77.54***[76.96-78.12] | 92.03** [91.35-92.70] |
| | EEGNet | 82.33***[81.98-82.69] | 75.62***[73.31-77.94] | 89.43 [87.00-91.86] |
| | ChronoNet | 84.35** [83.83-84.86] | 79.75** [79.02-80.48] | 89.21 [88.56-89.85] |
| | Multi-BK-Net | 85.95 [85.62-86.29] | 81.96 [81.39-82.53] | 90.18 [89.83-90.52] |
| male | Deep4Net | 83.25***[81.58-84.92] | 68.22***[63.93-72.51] | 97.05***[96.05-98.04] |
| (n=422) | ShallowNet | 85.28***[84.90-85.67] | 79.90***[78.50-81.30] | 90.23 [88.92-91.54] |
| | TCN | 85.43***[84.88-85.98] | 81.09***[80.41-81.76] | 89.41 [88.53-90.29] |
| | EEGNet | 84.19***[83.76-84.63] | 78.91***[76.38-81.44] | 89.05 [86.67-91.42] |
| | ChronoNet | 84.72***[84.03-85.40] | 82.03***[81.11-82.95] | 87.18* [86.29-88.07] |
| | Multi-BK-Net | 88.18 [87.78-88.57] | 86.98 [86.44-87.52] | 89.27 [88.34-90.20] |

### A.6.6 Training Time and Computational Requirements

We report training and inference time and memory in Table 20.

Table 20: Mean training and inference times across runs (n=10) in hours: minutes and mean CUDA memory across runs in GB. Training times are for training only, i.e., excluding data loading and pre-processing. These times are only intended to give a rough estimate of training and inference times. Inference time is based on 276 recordings of the TUAB evaluation set.

| Architecture | Training Time | Training Memory (GB) | Inference Time | Inference Memory (GB) |
|---|---|---|---|---|
| Deep4Net | 1:05 | 7.8 | 0:03 | 0.60 |
| ShallowNet | 1:06 | 4.01 | 0:05 | 10.30 |
| TCN | 2:08 | 2.88 | 0:02 | 0.51 |
| EEGNet | 1:06 | 2.61 | 0:02 | 0.82 |
| ChronoNet | 1:04 | 2.61 | 0:04 | 0.14 |
| Multi-BK-Net | 12:26 | 1.59 | 0:02 | 1.70 |

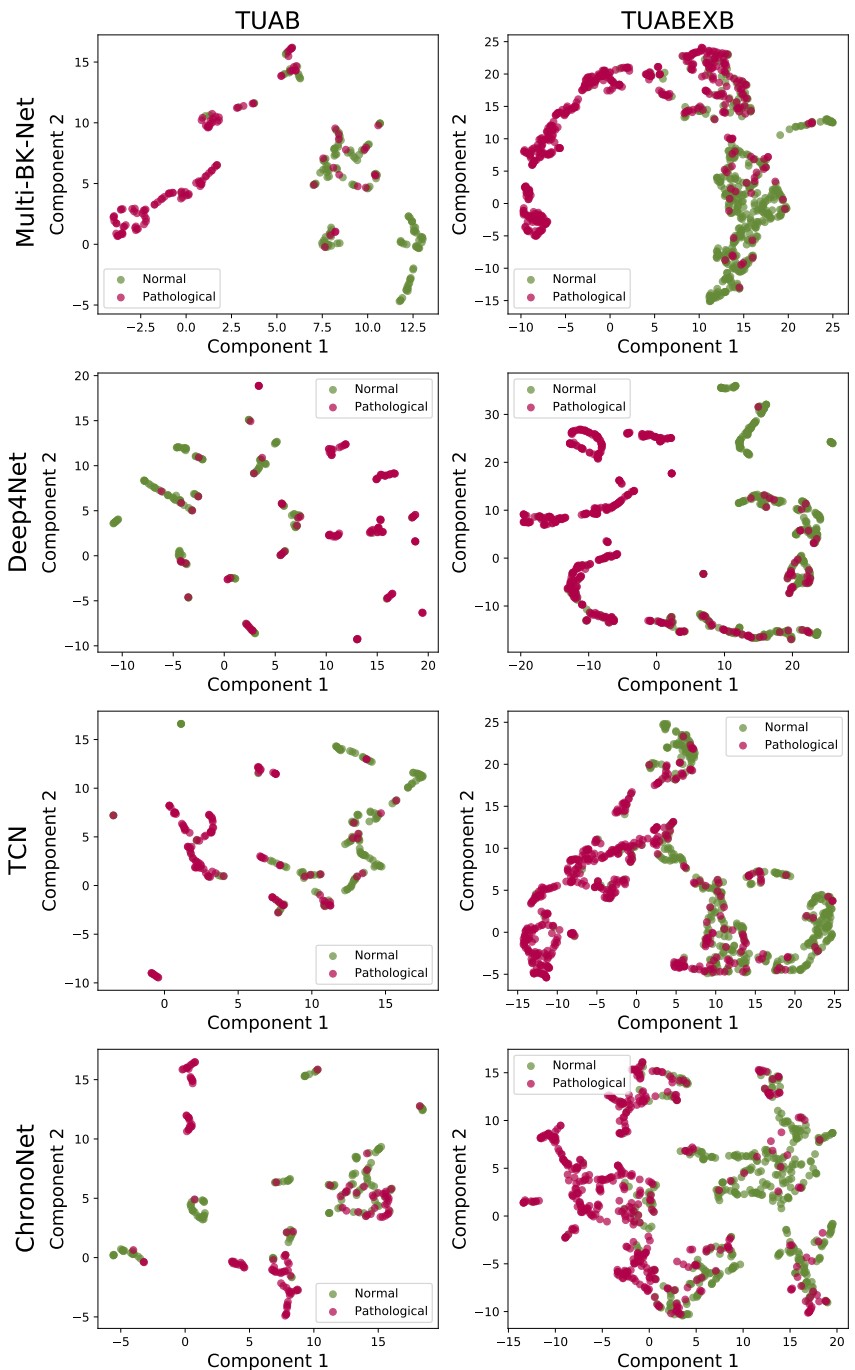

Figure 10: A comparison of UMAP visualisation of feature representations learned with Multi-BK-Net, Deep4Net, TCN and ChronoNet for pathological and non-pathological recordings on the predefined TUAB (left column) and the TUABEXB (right column) evaluation sets. Rows represent the UMAP visualisation of different models. Pink dots represent the pathological class, while green dots represent the non-pathological class.

