# OpenReview forum: "Multi-BK-Net: Multi-Branch Multi-Kernel Convolutional Neural Networks for Clinical EEG Analysis"
_TMLR — Accepted by TMLR_

### Review · Reviewer_nwjc · 2025-08-28

**Summary Of Contributions:**

The goal of this paper is to develop a more robust and generalizable deep learning model, Multi-BK-Net, for classifying pathological versus non-pathological EEG recordings by addressing data heterogeneity and intra-/inter-subject variability through a multi-branch, multi-kernel convolutional neural network architecture.
The paper introduces Multi-BK-Net, a multi-branch, multi-kernel convolutional neural network tailored for EEG pathology classification. The core innovation lies in the model’s first convolutional block, which contains five parallel branches, each with a distinct temporal kernel size corresponding to clinically relevant EEG frequency bands: delta, theta, alpha, beta, and low gamma. This design enables the network to capture both short- and long-term temporal patterns across a wide spectrum of physiological variability. Each branch performs temporal convolution followed by spatial convolution and mean pooling, with the outputs concatenated and processed through three additional convolutional blocks before final classification. The model was trained using data from two public EEG datasets—TUAB and TUABEXB—combined into a unified training set (TUABCOMB). To improve generalization and mitigate overfitting, training included trial-wise windowing (60 seconds each), group normalization, GELU activations, dropout regularization, and optimization via AdamW with cosine annealing. Hyperparameters were tuned using Optuna with multivariate Tree-structured Parzen Estimators. The evaluation used accuracy, sensitivity, F2-score, and Mann-Whitney U tests with Bonferroni correction to assess statistical significance, with additional interpretability provided through UMAP feature visualization and amplitude gradient analyses.

**Additional Comments:**

The paper has solid ideas, but the contribution is more associated to the novel application of already existing techniques.

**Audience:**

Yes

**Audience Explanation:**

EEG analysis is of interest to a broad audience in the TMLR community.

**Broader Impact Concerns:**

There is no broader impact, and one may be beneficial since the proposed model will be used for biomedical data analysis.

**Claims And Evidence:**

Yes

**Claims Explanation:**

The paper evaluates Multi-BK-Net across two EEG datasets (TUAB and TUABEXB), including both a standard benchmark and a more diverse dataset, and compares its performance to five baseline CNNs and multiple published state-of-the-art models. This dual-dataset evaluation strengthens the credibility of its claims and benchmarks. The paper implements robust evaluation practices including 10-run averages, multiple metrics (accuracy, sensitivity, F2-score), Bonferroni-corrected Mann–Whitney U tests, and ablation studies to isolate architectural contributions. This rigor addresses concerns about variance and reproducibility common in deep learning research.
The paper goes beyond performance metrics to include UMAP visualizations and amplitude gradient analysis. These interpretability tools demonstrate that the learned representations align with known neurophysiological features of pathological EEGs, thereby increasing the model’s trustworthiness for clinical applications.

That being said, some points could be improved. First, single experts label the datasets used, and the authors acknowledge that inter-rater variability may cap classification accuracy. However, no attempt is made to integrate consensus labels or probabilistic (fuzzy) targets that could mitigate this issue. Second, while the model performs well on public datasets, the paper does not include validation in a real-world clinical setting (e.g., clinician feedback or pilot deployment), which limits conclusions about practical adoption or usability.

**Requested Changes:**

I would suggest the authors to work on the two weaknesses listed above. I also have a question that could help clarify the contribution:
Can the use of multi-rater annotations or fuzzy labels improve model performance and better reflect diagnostic uncertainty in EEG interpretation?

---

> ### Author Response · Authors · 2025-10-27
> **Response to Reviewer nwjc Part 1**
>
> Thank you for your review. Your comments are helping us to improve our work. Below, we will address each of the points you raised:
>
> **1.“ First, single experts label the datasets used, and the authors acknowledge that inter-rater variability may cap classification accuracy. However, no attempt is made to integrate consensus labels or probabilistic (fuzzy) targets that could mitigate this issue.”**
>
> We agree that both multi-rater annotations and fuzzy labels are valuable strategies. Previous studies in deep learning-based medical EEG analysis have also shown this. Multi-rater annotations help build more robust, expert-aligned models by capturing the full scope of human interpretation. Fuzzy labels provide a mathematical framework for intrinsically modelling the ambiguity and inherent uncertainty in EEG signals, leading to more nuanced predictions and potentially greater accuracy, especially in clinically challenging scenarios. Together, these approaches contribute to creating more reliable and clinically relevant deep learning-based systems for clinical EEG analysis.
> Unfortunately, to date, only single-rater labels are available for general EEG pathology datasets. Therefore, providing multi-rater annotations is a crucial step for advancing deep learning-based general EEG pathology classification. It marks a significant beginning in improving label quality and is also required for clinical adoption. However, EEG analysis by human experts is time-consuming. For example, Brogger et al. (2018) have reported a median time of 12.5 minutes to review a 20-minute EEG.
> In contrast, the review time can be longer for abnormal EEG (median 20.7 minutes), for younger patients, or for EEG without sleep (Brogger et al. 2018). The datasets used in our study together contain 11,872 EEG recordings totalling at least 2968 hours. Providing multi-rater annotations for this amount of EEG data would take several months to years and require considerable financial resources. Therefore, we cannot integrate this into the current study. As a result, incorporating consensus labels or fuzzy labels is also not yet possible. However, future studies focusing on clinician feedback or pilot deployment could include fuzzy labels, which is also a crucial step for advancing and clinically adopting deep learning-based support systems, as you have thankfully pointed out. Therefore, we revised the discussion and conclusion (Section 4) to include a more detailed paragraph on the impact of multi-rater annotations and fuzzy labels on classification accuracy, reflecting the topic's importance.

---

> > ### Author Response · Authors · 2025-10-27
> > **Response to Reviewer nwjc Part 2**
> >
> > **2)“Second, while the model performs well on public datasets, the paper does not include validation in a real-world clinical setting (e.g., clinician feedback or pilot deployment), which limits conclusions about practical adoption or usability.”**
> >
> > Thank you for pointing out this issue. We agree that validation in a real-world clinical setting is an important step to evaluate the reliability of deep learning-based methods for general EEG pathology classification before clinical application. Therefore, a more comprehensive exploration of these areas is required to improve reliability in real-world settings before deep learning-based (support) systems for the classification of EEG pathologies can be used in everyday clinical practice. While real-world application testing is undeniably crucial for validating deep learning methods, leveraging public datasets like the TUH EEG Corpus (TUEG) (Obeid & Picone, 2016) for initial development and evaluation offers significant advantages. The TUEG serves as a valid foundation dataset for several reasons:
> > First, the TUEG contains clinical EEG data that exhibit wide variation across essential parameters, such as patient age, diagnoses, and medications. This inherent diversity in terms of volume and heterogeneity is one essential characteristic. These are critical factors for training robust machine learning models that can represent and perform effectively in real-world settings. Secondly, the EEG recordings within the TUEG have been collected directly in a clinical setting. This contrasts with tightly controlled research environments, which are often less variable. A clinical setting inherently introduces greater variability in parameters such as electrode location, the clinical environment, equipment variations, and noise levels. Capturing this variability is crucial for the development of high-performance and robust support systems.  Finally, the TUEG is an ongoing data collection effort that already contains 14 years of clinical EEG data collected at Temple University Hospital. The size and ongoing nature of this resource provide researchers with an exceptional opportunity to develop, train, and test deep learning models on datasets that closely mirror the complexities and challenges encountered in everyday clinical practice. Therefore, by utilising the TUEG dataset, researchers can take a vital first step in developing deep learning methods that represent the variable nature of EEG data for real-world clinical applications.
> > Nevertheless, real-world application testing is undeniably crucial for validating deep learning methods, and is an essential step for translation into everyday clinical practice. In our study, we focus on the initial evaluation of the effectiveness of multi-scale,  multi-branch CNNs for general EEG pathology classification. Addressing these specific questions in real-world applications requires more thorough studies using clinician feedback or pilot deployment and is thus unfortunately outside the scope of our current work. Therefore, we revised the discussion (Section 4) to include real-world application testing as a recommendation for future research work.
> >
> > Thank you again for your constructive feedback.
> >
> > ### References:
> > Brogger, J., Eichele, T., Aanestad, E., Olberg, H., Hjelland, I., & Aurlien, H. (2018). Visual EEG reviewing times with SCORE EEG. Clinical neurophysiology practice, 3, 59-64.
> >
> > Obeid, I., & Picone, J. (2016). The temple university hospital EEG data corpus. Frontiers in neuroscience, 10, 196.

---

### Review · Reviewer_NCdX · 2025-09-29

**Summary Of Contributions:**

This paper presents Multi-BK-Net, a convolutional neural network designed to classify EEG recordings as pathological or normal by employing multiple kernel sizes that explicitly target different frequency bands. The central idea is to align convolutional kernels with conventional EEG frequency ranges, with the goal of enhancing generalization across heterogeneous signals. Experiments on the TUAB and TUABEXB datasets show modest improvements over baselines. The authors further support their approach with ablation studies and interpretability analyses.

**Strengths**

* The paper is well-written and well-structured, with clear motivation and thorough experimentation.
* The evaluation is comprehensive: the proposed method is compared against multiple baselines, and several ablation studies are included.
* The study uses a cross-subject design, ensuring no patient overlap between training and test data.

**Weaknesses**

*Incremental novelty: Using CNNs to capture different frequency bands is not novel. The main contribution lies in tailoring kernel sizes to EEG frequency bands, which is an incremental improvement. While the results are statistically significant, the actual performance gain is modest, about 2% over the baseline and less than 1% compared to single-branch or single-kernel ablations. It remains unclear what clinical value such improvements provide.

* Although TUABEXB is larger and more diverse, both datasets originate from the same institution, raising concerns about cross-institutional generalization.

**Audience:**

Yes

**Audience Explanation:**

Maybe. People in the clinical EEG field will read this.

**Claims And Evidence:**

No

**Claims Explanation:**

The marginal improvement limits the potential impact of this paper to the community.

**Requested Changes:**

Please clarify your statistical test, it it on n=5 repeated runs?

---

> ### Author Response · Authors · 2025-10-27
> **Response to Reviewer NCdX Part 1**
>
> We sincerely thank you for your valuable feedback. We are happy that you found our manuscript “well-written” and “well-structured” and that our “evaluation is comprehensive”. We appreciate your feedback and help in improving the clarity of our work.  In the following, we will address the mentioned weaknesses and respond to your concerns.
>
> **Weaknesses:
> Incremental novelty: Using CNNs to capture different frequency bands is not novel. The main contribution lies in tailoring kernel sizes to EEG frequency bands, which is an incremental improvement. While the results are statistically significant, the actual performance gain is modest, about 2% over the baseline and less than 1% compared to single-branch or single-kernel ablations. It remains unclear what clinical value such improvements provide.**
>
> Research has shown that CNNs have been quite effective for general EEG pathology classification due to their ability to extract relevant feature representations directly from raw or minimally preprocessed EEG data (Khan et al., 2022; Schirrmeister et al., 2017; Roy et al., 2019; Wu et al., 2021). Despite these significant advancements, three types of limitations and challenges persist: scarcity of large-scale training data, limited model capability to handle EEG heterogeneity, and irreducible errors imposed by various factors, including label noise.
>
> In this study, we focus on the limited ability of current CNNs to handle EEG heterogeneity by developing a CNN that extracts features across multiple frequency/time ranges, thereby providing a more complete representation of EEG signals. Multi-scale and multi-branch CNNs are specifically designed to tackle several key challenges in deep learning-based EEG classification, with data heterogeneity being a significant one (Ko et al. 2021).
> By achieving a mean accuracy of 87% on the larger, diverse TUABEXB, our Multi-BK-Net is setting a new benchmark on this dataset.  In addition, with mean accuracies in the narrow range of 87–88%, the Multi-BK-Net approaches the upper limit of inter-rater agreement among human experts (Beuchat et al., 2021), thus bringing deep learning-based general EEG pathology classification closer to human-level performance.
> An improvement in evaluation sensitivity demonstrates further practical gains of our method.
> In deep learning-based clinical EEG analysis, high accuracy and sensitivity are of greatest importance because they directly translate to improved patient outcomes, reduced diagnostic errors, and enhanced clinical efficiency.  High sensitivity ensures that critical events, such as seizures, are not missed, preventing potential patient harm and delays in treatment (Iešmantas & Alzbutas, 2020; Wang et al., 2023). Accuracy is crucial for reliable diagnosis and proper patient management. Even minor improvements in these metrics are valuable because they can significantly reduce false detections, achieve expert-level agreement, and expedite physician analysis, making the systems more trustworthy and practical for everyday clinical use (Hogan et al.,2025; Saab et al., 2020). Such advancements can also lead to better trade-offs between sensitivity and false-positive rates, allowing clinicians to tune the system for optimal performance in specific scenarios (Wang et al., 2023).
> Considering this, demonstrating the effectiveness of our Multi-BK-Net on a large heterogeneous dataset constitutes a significant and relevant contribution. We rephrased the contribution paragraph (Section 1) to include more detail on the clinical value that these improvements provide. In addition, we included a paragraph discussing the clinical importance of even minor improvements in accuracy and sensitivity in Section 4 (Discussion and Conclusion).

---

> > ### Author Response · Authors · 2025-10-27
> > **Response to Reviewer NCdX Part 2**
> >
> > **Although TUABEXB is larger and more diverse, both datasets originate from the same institution, raising concerns about cross-institutional generalization.**
> >
> > We fully agree with you.  Evaluating cross-dataset generalisation performance is a crucial topic that warrants closer examination.
> > At the same time, to date, the possibilities for empirically investigating generalisation performance across multiple datasets remain limited.  Currently, there are only two open-source datasets for EEG pathology classification that are not derived from the TUEG: the CAUEEG (Kim et al. 2023) and the  NMT Scalp EEG Dataset (Khan et al. 2022). Unfortunately, as the CAUEEG contains fewer than 21 EEG channels, we cannot employ the models used in our study, as they are designed, optimised, and trained for an input consisting of 21 EEG channels. Therefore, only NMT can be considered in our framework. The NMT contains 2,417 recordings collected from a South-Asian demographic.
> > Khan et al. (2022) have already evaluated the generalisation performance of  Deep4Net and ShallowNet on the NMT after training them on the TUAB dataset and observed a degradation in performance (48-50% accuracy), indicating a distribution shift. In addition, the authors discussed several factors that potentially contributed to this cross-dataset variability.  Furthermore, Khan et al. (2022) demonstrated that fine-tuning can mitigate the effects of cross-dataset variability. Similar findings have been observed by Darvishi-Bayazi et al. (2024). In addition, the authors explored several cross-dataset transfer learning approaches. They identified the challenges of negative transfer and emphasised the importance of certain components in overcoming distribution shifts, potential spurious correlations, and achieving positive transfer.  Therefore, we have not replicated this analysis on cross-dataset transfer learning approaches.
> >
> > However, we evaluated the generalisation performance of our proposed Multi-BK-Net and our five baseline approaches trained on the TUH on the predefined NMT evaluation and included this analysis in Section 3.10. As discussed in previous work (Khan et al. 2022; Darvishi-Bayazi et al. 2024), we also observed performance degradation across all models, but less so for our proposed Multi-BK-Net than for the baseline CNNs.
> >
> > Besides the fact that cross-dataset performance between the TUH dataset and NMT has already been preliminarily explored, and a complete analysis of multiple datasets is not currently possible due to the lack of other datasets, it would also require the application of transfer learning techniques, as demonstrated by Darvishi-Bayazi et al. (2024). This would therefore exceed the scope of this study.  Although the present work focused on data from a single centre, cross-centre validation is an important future direction. We additionally included a comment in our discussion (Section 4) suggesting cross-centre studies to estimate generalisation error as a future research topic.
> >
> > **Requested Changes: Please clarify your statistical test, is it on n=5 repeated runs?**
> >
> > We apologise for the lack of clarity. For each architecture, we repeated training 10 times with different random seeds, i.e., 10 independent runs. We reported the average of evaluation metrics across multiple runs (N=10) using different random seeds. Thus, the averaged results and statistical tests are based on n=10 repeated runs. We have clarified this point in  Section 3.3 (“Evaluation of Classification Performance”). In addition, we have included this information in the captions of the corresponding figures (Figures 2, 3, 4, 5, 8, and 9) and tables (Tables 2, 3, 4, 5, 6, 7, 8, 18, 19, and 20).

---

> > > ### Author Response · Authors · 2025-10-27
> > > **Response to Reviewer NCdX Part 3**
> > >
> > > ### References:
> > >
> > > Beuchat, I., Alloussi, S., Reif, P. S., Sterlepper, N., Rosenow, F., & Strzelczyk, A. (2021). Prospective evaluation of interrater agreement between EEG technologists and neurophysiologists. Scientific reports, 11(1), 13406.
> > >
> > > Darvishi-Bayazi, M. J., Ghaemi, M. S., Lesort, T., Arefin, M. R., Faubert, J., & Rish, I. (2024). Amplifying pathological detection in EEG signaling pathways through cross-dataset transfer learning. Computers in biology and medicine, 169, 107893.
> > >
> > > Iešmantas, T., & Alzbutas, R. (2020). Convolutional neural network for detection and classification of seizures in clinical data. Medical & Biological Engineering & Computing, 58(9), 1919.
> > >
> > > Hogan, R. C., Mathieson, S., Luca, A., Ventura, S., Griffin, S., Boylan, G. B., & O’Toole, J. M. (2025). Scaling convolutional neural networks achieves expert level seizure detection in neonatal EEG. Npj Digital Medicine, 8(1).
> > >
> > > Khan, H. A., Ul Ain, R., Kamboh, A. M., Butt, H. T., Shafait, S., Alamgir, W., ... & Shafait, F. (2022). The NMT scalp EEG dataset: An open-source annotated dataset of healthy and pathological EEG recordings for predictive modeling. Frontiers in neuroscience, 15, 755817.
> > > Kim, M. J., Youn, Y. C., & Paik, J. (2023). Deep learning-based EEG analysis to classify normal, mild cognitive impairment, and dementia: Algorithms and dataset. NeuroImage, 272, 120054.
> > > Ko, W., Jeon, E., Jeong, S., & Suk, H. I. (2021). Multi-scale neural network for EEG representation learning in BCI. IEEE Computational Intelligence Magazine, 16(2), 31-45.
> > >
> > > Roy, Y., Banville, H., Albuquerque, I., Gramfort, A., Falk, T. H., & Faubert, J. (2019). Deep learning-based electroencephalography analysis: a systematic review. Journal of neural engineering, 16(5), 051001.
> > >
> > > Saab, K., Dunnmon, J., Ré, C., Rubin, D. L., & Lee‐Messer, C. (2020). Weak supervision as an efficient approach for automated seizure detection in electroencephalography. Npj Digital Medicine, 3(1).
> > >
> > > Schirrmeister, R., Gemein, L., Eggensperger, K., Hutter, F., & Ball, T. (2017). Deep learning with convolutional neural networks for decoding and visualization of eeg pathology. arXiv e-prints, arXiv-1708.
> > >
> > >
> > > Wang, X., Wang, X., Wang, C., Wang, Z., Liu, X., Lv, X., & Tang, Y. (2023). A Two-Stage Automatic System for Detection of Interictal Epileptiform Discharges from Scalp Electroencephalograms. eNeuro, 10(11).
> > >
> > > Wu, T., Kong, X., Wang, Y., Yang, X., Liu, J., & Qi, J. (2021, July). Automatic classification of EEG signals via deep learning. In 2021 IEEE 19th International Conference on Industrial Informatics (INDIN) (pp. 1-6). IEEE.

---

### Review · Reviewer_ESqg · 2025-10-06

**Summary Of Contributions:**

The authors propose a multibranch, multikernel convolutional neural network, named Multi-BK-Net, to classify clinical EEGs as pathological or nonpathological. They designed the model with an initial block containing five parallel temporal kernels, each targeting a specific clinical frequency band from delta to low gamma.
The authors trained their model on the TUABCOMB and TUABEXB datasets and evaluated it on the respective test splits. They report accuracy and sensitivity improvements over baseline methods.
The authors' study has some limitations. They tested the model only on datasets from a single institution (TUH), which can limit its validity. Some other limitations: missing ROC curves; address potential data leakage from combining the training sets; limited analysis of potential biases due to factors like age or sex.

**Additional Comments:**

None

**Audience:**

Yes

**Audience Explanation:**

The authors position their paper as “machine learning for EEG pathology screening in healthcare”. They offer a pragmatic CNN based setup that improves sensitivity on EEG based pathology detection. The authors highlight the model's design and implementation as a valuable contribution for both machine learning and neuroengineering audiences.

**Broader Impact Concerns:**

The authors can include discussing the risk of automation bias and missed pathology if the model is deployed without calibrated thresholds and essential human in the loop guardrails. The paper should acknowledge the potential for institutional bias from single site training and the risk of demographic performance disparities. It should also address clinical challenges such as data governance and handling (of medical datasets) and the mitigation of model drift over time.

**Claims And Evidence:**

Yes

**Claims Explanation:**

The authors show that their results outperform baselines on two predefined test sets. They support their claims with multiple seeds and ablation studies that isolate the contribution of their first-block design. They also report the consistency of the interpretability analyses.
However, the authors' conclusions about "robust generalisation" may be limited, as they only demonstrated this within the TUH corpora.

**Requested Changes:**

Some suggested enhancements:
1) External Validation: Evaluate the model on at least one external, non-TUH dataset to substantiate claims of generalisation.
2) Data Leakage: Address if patient or recording overlaps between the training and test datasets for all datasets.
3) Clinical Metrics: Report ROC/PR curves and AUCs. For metrics comparison please  provide sensitivity at a fixed 95% specificity for clinical relevance (this is a better metric to compare ML models).
4) Robustness Analysis: Subgroup analyses for age, sex, artifacts, including confidence intervals for each group would be benificial.

Would strengthen the work
 5) Compute/latency: Report training/inference memory, time, and real-time feasibility.
 6) Error analysis: Qualitative review of false negatives/positives for EEGs and report alignment with clinical notes.

---

> ### Author Response · Authors · 2025-10-27
> **Response to Reviewer ESqg Part 1**
>
> Thank you for your thoughtful feedback and recommendations to improve the quality of our paper. In the following, we will address the mentioned concerns and requested changes.
>
> **1)“External Validation: Evaluate the model on at least one external, non-TUH dataset to substantiate claims of generalisation.”:**
>
> We agree and thank you for pointing this out.  Evaluating the cross-dataset generalisation performance is a crucial topic that warrants closer examination. At the same time, the possibilities for empirically investigating generalisation performance across multiple datasets remain limited to date. Currently, there are only two open-source datasets for EEG pathology classification that are not derived from the TUEG: the CAUEEG (Kim et al. 2023) and the  NMT Scalp EEG Dataset (Khan et al. 2022). Unfortunately, as the CAUEEG contains fewer than 21 EEG channels, we cannot employ the models used in our study, as they are designed, optimised, and trained for an input consisting of 21 EEG channels. Therefore, only NMT is considered in our framework. The NMT contains 2,417 recordings collected from a South-Asian demographic. In agreement with reviewer NCdX’s comment, we have also included cross-dataset performance results for our models on another external dataset. In particular, we evaluated the generalisation performance of the Multi-BK-Net and the five baseline models, which were trained on the TUH dataset, on the predefined NMT evaluation set in Section 3.10. As discussed in previous work (Khan et al. 2022; Darvishi-Bayazi et al. 2024), we also observed performance degradation across all models, but less so for our proposed Multi-BK-Net than for the baseline CNNs.
>
> **2)“Data Leakage: Address if patient or recording overlaps between the training and test datasets for all datasets.”**
>
> We thank you for your comment and apologise for the confusion. There is no data leakage. We have clarified this in Section 3.1 “Datasets and preprocessing steps”, where we wrote “There is no overlap of patients between the TAUB and the TUABEXB datasets. In both datasets, there is also no overlap of patients between the predefined training set and the final evaluation set.”
>
> **3)“Clinical Metrics: Report ROC/PR curves and AUCs. For metrics comparison, please provide sensitivity at a fixed 95% specificity for clinical relevance (this is a better metric to compare ML models).”**
>
> We thank you for pointing this out. We agree and have included the mean ROC/PR curves and AUCs for the Multi-BK-Net and all five baseline approaches on the TUAB and TUABEXB datasets in Section 3.4 and Figures 4 and 5. In addition, for each architecture and test set, we included the mean sensitivities averaged across 10 runs at a fixed 95% specificity for clinical relevance in Table 2, Section 3.4.
>
> **4)“Robustness Analysis: Subgroup analyses for age, sex, and artefacts, including confidence intervals for each group, would be beneficial.”**
>
> We thank you for this comment. Information about age and sex is provided in the header of each EEG recording and can be used for further analysis. Unfortunately, no information about artefacts is given in the EEG data. While a detailed analysis of EEG artefact subgroups would offer valuable insights into model performance, such an analysis poses significant methodological challenges that should be addressed in future work. We included subgroup analyses by age and sex in Section 3.4 and in Tables 18 and 19 (Appendix A.6.5).
>
> **5) Compute/latency: Report training/inference memory, time, and real-time feasibility.**
>
> We report  the training/inference memory, time in Table 20 (Appendix A.6.6).
>
> **6) Error analysis: Qualitative review of false negatives/positives for EEGs and report alignment with clinical notes.**
>
> We agree with you.  A qualitative review of false negatives and positives in EEG, aligned with clinical notes, could provide more clinically valuable insights and explanations. We therefore included a qualitative analysis of false negatives/positives of our Multi-BK-Net and report alignment with clinical notes in Section 3.9.

---

> > ### Author Response · Authors · 2025-10-27
> > **Response to Reviewer ESqg Part 2**
> >
> > **Broader Impact Concerns: The authors can include discussing the risk of automation bias and missed pathology if the model is deployed without calibrated thresholds and essential human-in-the-loop guardrails. The paper should acknowledge the potential for institutional bias from single-site training and the risk of demographic performance disparities. It should also address clinical challenges such as data governance and handling (of medical datasets) and the mitigation of model drift over time.**
> >
> > Thank you for pointing out this issue. We agree that we should address the risk of automation bias and missed pathology, acknowledge the potential for institutional bias from single-site training, and the risk of demographic performance disparities. Additionally, we should address clinical challenges such as data governance and handling (of medical datasets) and the mitigation of model drift over time. Therefore, we have added extra paragraphs to Section 4 to discuss these risks and challenges.
> >
> > ### References:
> > Kim, M. J., Youn, Y. C., & Paik, J. (2023). Deep learning-based EEG analysis to classify normal, mild cognitive impairment, and dementia: Algorithms and dataset. NeuroImage, 272, 120054.
> >
> > Khan, H. A., Ul Ain, R., Kamboh, A. M., Butt, H. T., Shafait, S., Alamgir, W., ... & Shafait, F. (2022). The NMT scalp EEG dataset: An open-source annotated dataset of healthy and pathological EEG recordings for predictive modeling. Frontiers in neuroscience, 15, 755817.

---

### Author Response · Authors · 2025-10-27
**Notification of Revised Manuscript Submission**

Dear Reviewers,
We have uploaded a revised version of the manuscript, incorporating your valuable suggestions. The modifications are highlighted in **blue** for your convenience. Additionally, please note that there are changes in the appendix. We hope these updates address all the remaining concerns raised in your reviews.

---

### Decision · Action_Editor_22qB · 2025-12-02

**Recommendation:** Accept with minor revision

**Additional Comments:**

After the authors' revisions, all reviewers updated their stance to Leaning Accept, indicating that their major concerns were addressed. Remaining comments are minor and mostly relate to phrasing, such as:
- Avoid overclaiming “generalization” in the abstract and discussion
- Clarify limitations of single-institution datasets
- Maintain consistency in how performance limitations are framed

**Audience:**

Yes

**Audience Explanation:**

All reviewers agree that this work is relevant and interesting to TMLR’s community. The paper sits at the intersection of:
- Machine learning for healthcare
- Deep learning for EEG analysis
- Clinical decision-support research
These areas have strong readership within TMLR. The model's design (multi-branch/multi-kernel CNN aligned with EEG frequency bands) and its application to large public clinical datasets make it relevant to ML researchers, biomedical signal processing experts, and neuroengineering practitioners.

**Claims And Evidence:**

Yes

**Claims Explanation:**

Across all three reviewers, there is agreement that the paper’s claims are supported by empirical evidence. The authors provide:
- Benchmarks against multiple baselines
- Multi-run evaluation
- Ablation studies isolating the multi-branch/multi-kernel contributions
- ROC/PR curves, AUCs, and clinically relevant sensitivity/specificity metrics
- Subgroup analyses (age/sex)
- External validation on the NMT dataset
Reviewers noted that while the generalization claims should be phrased more cautiously (because cross-institution performance still drops), the evidence presented is rigorous, transparent, and convincingly supports the core technical claims. One reviewer originally said “No” due to modest effect size and incremental novelty, but even that reviewer acknowledged the results are statistically significant and well-supported. Overall, the evidence quality meets TMLR standards.